# Amino acid-dependent cMyc expression is essential for NK cell metabolic and functional responses in mice

Róisín M. Loftus[1], Nadine Assmann[1], Nidhi Kedia-Mehta[1], Katie L. O'Brien[1], Arianne Garcia[1], Conor Gillespie[1], Jens L. Hukelmann[2,3], Peter J. Oefner[4], Angus I. Lamond[2], Clair M. Gardiner[1], Katja Dettmer [4], Doreen A. Cantrell [3], Linda V. Sinclair [3] & David K. Finlay [1,5]

Natural killer (NK) cells are lymphocytes with important anti-tumour functions. Cytokine activation of NK cell glycolysis and oxidative phosphorylation (OXPHOS) are essential for robust NK cell responses. However, the mechanisms leading to this metabolic phenotype are unclear. Here we show that the transcription factor cMyc is essential for IL-2/IL-12-induced metabolic and functional responses in mice. cMyc protein levels are acutely regulated by amino acids; cMyc protein is lost rapidly when glutamine is withdrawn or when system L-amino acid transport is blocked. We identify SLC7A5 as the predominant system L-amino acid transporter in activated NK cells. Unlike other lymphocyte subsets, glutaminolysis and the tricarboxylic acid cycle do not sustain OXPHOS in activated NK cells. Glutamine withdrawal, but not the inhibition of glutaminolysis, results in the loss of cMyc protein, reduced cell growth and impaired NK cell responses. These data identify an essential role for amino acid-controlled cMyc for NK cell metabolism and function.

[1] School of Biochemistry and Immunology, Trinity Biomedical Sciences Institute, Trinity College Dublin, 152-160 Pearse Street, Dublin 2, Ireland. [2] Centre for Gene Regulation and Expression, School of Life Sciences, University of Dundee, Dow Street, DD1 5EH, Scotland, UK. [3] Division of Cell Signalling and Immunology, School of Life Sciences, University of Dundee, Dow Street, DD1 5EH Scotland, UK. [4] Institute of Functional Genomics, University of Regensburg, 93053 Regensburg, Germany. [5] School of Pharmacy and Pharmaceutical Sciences, Trinity Biomedical Sciences Institute, Trinity College Dublin, 152-160 Pearse Street, Dublin 2, Ireland. Correspondence and requests for materials should be addressed to D.K.F. (email: finlayd@tcd.ie)

Natural killer (NK) cells are important effector lymphocytes for anti-tumour and anti-viral immune responses. Activated NK cells undergo substantial changes in cellular metabolic pathways, undergoing reprogramming to achieve increased rates of glycolysis and mitochondrial oxidative phosphorylation (OXPHOS)[1–3]. Elevated glucose metabolism is a common feature of many activated immune cells and is required to provide the energy and the biosynthetic capacity to sustain immune functions[4]. Glucose is metabolised to pyruvate by glycolysis and then either converted to lactate, which is secreted from the cell, or further metabolised within the mitochondria to fuel OXPHOS. The amino acid glutamine is also an important fuel for metabolically active cells as glutaminolysis feeds into the tricarboxylic acid cycle (TCA) to fuel OXPHOS. Our previous research has shown that the changes in glucose metabolism that occur during NK cell activation are crucial for NK cell functional responses, including the production of interferon-γ (IFNγ) and the expression of the cytotoxic molecule granzyme B[1–3]. This research provides important insights into why NK cells may be dysfunctional within solid tumours[5–7], where the microenvironment contains low levels of glucose that would curtail NK cell metabolism[8,9]. Although NK cell-based cancer immunotherapies have had success in the treatment of haematological malignancies, the efficacy of these approaches has been less successful for solid tumours[10]. Understanding how the nutrient-restrictive tumour microenvironment affects NK cell metabolism and function is crucial to developing new strategies that induce robust NK cell anti-cancer responses.

Although it is now clear that glucose metabolism is important in the control of NK cell responses, the mechanisms involved are unclear. The mammalian target of rapamycin complex 1 (mTORC1) is an important regulator of immune responses that has well-described functions in the control of cellular metabolism in multiple immune subsets[4]. In NK cells, mTORC1 is required for the induction of elevated glycolysis following cytokine stimulation[1,3,11]. In T-cell populations, the transcription factors hypoxia-inducible factor-1α (HIF1α) and cMyc have been described as central glycolytic regulators[12–14]. HIF1α is an important transcriptional regulator of the cellular response under hypoxic conditions, but can also be expressed under normoxic conditions in which it has an important function in controlling immune responses. HIF1α regulates glycolytic responses in multiple T-cell subsets, including interleukin-2 (IL-2)-cultured CD8+ cytotoxic T lymphocytes (CTLs), by promoting the expression of glucose transporters and glycolytic genes[12,15]. In T cells, the transcription factor cMyc controls the early metabolic reprogramming events that occur following T-cell receptor (TCR) activation by increasing the expression of glucose transporters, glycolytic enzymes and enzymes involved in glutaminolysis[14]. cMyc has also been implicated in the control of invariant NKT cell development in the thymus[16]. However, nothing is currently known about the role of HIF1α and cMyc in NK cell metabolic or functional responses.

Elevated OXPHOS is also essential for NK cell functional responses, but little is known regarding the mechanisms involved in the induction of mitochondrial metabolism in cytokine-activated NK cells[3,17]. Glutamine is an important fuel source for sustaining mitochondrial OXPHOS in activated T cells, but whether glutamine is an important fuel for NK OXPHOS has not be studied[14].

Herein, we show that cMyc expression is crucial for NK cell metabolic and functional responses. We identify mechanisms that control cMyc in NK cells, highlighting an important function for amino acid transport through SLC7A5 in regulating cMyc protein expression. Furthermore, these data show that cMyc protein expression is acutely sensitive to the availability of glutamine. We demonstrate that although glutamine does feed into the TCA cycle through glutaminolysis, this glutamine-fuelled TCA cycle is not important for sustaining elevated levels of OXPHOS in activated NK cells. Furthermore, we identify an important function for glutamine in NK cells; glutamine-regulated cMyc expression acts as a critical metabolic rheostat in controlling NK cell growth and effector responses. This study suggests that therapeutic strategies that stabilise cMyc expression in NK cells will lead to enhanced anti-tumour responses.

## Results

**cMyc controls NK cell metabolic and functional responses.** We have previously shown that NK cells undergo robust metabolic reprogramming in response to cytokine stimulation[1]. As the transcription factors cMyc and HIF1α have both been described to have an important role in promoting glycolytic metabolism in other lymphocyte subsets, we considered whether cMyc or HIF1α are required for IL-2/IL-12-induced NK cell metabolism and function[12,13,18–20]. To obtain the cell numbers required for biochemical analyses, splenic NK cells were expanded in low-dose IL-15 for 6 days, as previously described[1,2]. 'NK cells' refers to IL-15-expanded NK cells herein unless otherwise stated. cMyc protein expression was substantially increased in NK cells stimulated with IL-2/IL-12 for 18 h (Fig. 1a). In fact, IL-2/IL-12 stimulation induced cMyc protein levels rapidly within 30 min of stimulation (Fig. 1b). In contrast, HIF1α protein levels are not robustly induced in IL-2/IL-12-stimulated NK cells (Fig. 1c). As a positive control for HIF1α, NK cells were treated with dimethyloxalylglycine (DMOG) for the final 2 h prior to lysis; DMOG inhibits the prolyl-hydroxylases that target HIF1α for degradation resulting in the accumulation of HIF1α protein (Fig. 1c). The relative protein levels of cMyc and HIF1α were confirmed using a quantitative proteomics approach. The data showed eightfold more cMyc protein compared to HIF1α in activated NK cells (Fig. 1d). To determine whether IL-2/IL-12-induced cMyc is required for NK cell metabolic responses, $Myc^{flox/flox}$ × Tamox-Cre mice were generated to facilitate the inducible, cre recombinase-mediated excision of exon 2 of the Myc gene. cMyc-null ($Myc^{-/-}$) NK cells were generated by treating NK cells, cultured from splenocytes, with 4-hydroxytamoxifen and cMyc deletion was confirmed in each experiment (Fig. 1e). The wild-type controls (WT) were NK cells from $Myc^{WT/WT}$ Tamox-Cre mice treated equivalently with 4-hydroxytamoxifen. There was no difference in NK cell subsets in $Myc^{-/-}$ vs. WT cultures based on the expression of CD27/CD11b (Supplementary Fig. 1a). Unstimulated NK cells did not express appreciable levels of cMyc protein (Fig. 1a, b). Unstimulated $Myc^{-/-}$ NK cells were equivalent in size, as measured by forward scatter (FSC-A), to WT controls and neither WT controls nor $Myc^{-/-}$ NK cells express the transferrin receptor CD71, a known cMyc target gene (Fig. 1f, Supplemental Fig. 1b)[21]. Following IL-2/IL-12 stimulation, WT NK cells undergo blastogenesis, which involves an increase in the expression of nutrient receptors, including SLC2A1 (also called Glut1) and CD71 and a substantial increase in cell size, as measured by forward scatter (FSC-A) (Fig. 1g)[1]. In contrast, IL-2/IL-12-stimulated $Myc^{-/-}$ NK cells were small, with comparable in FSC-A to unstimulated NK cells, and expressed low levels of CD71 (Fig. 1g). Metabolic flux analysis showed significantly decreased rates of glycolysis and reduced glycolytic capacity in IL-2/IL-12-stimulated $Myc^{-/-}$ NK cells as compared to WT NK cells (Fig. 1h, Supplementary Fig. 1c), which was associated with decreased messenger RNA (mRNA) expression of the glycolytic machinery (Fig. 1i). IL-2/IL-12 stimulated $Myc^{-/-}$ NK cells also had reduced levels of OXPHOS and reduced maximal respiration

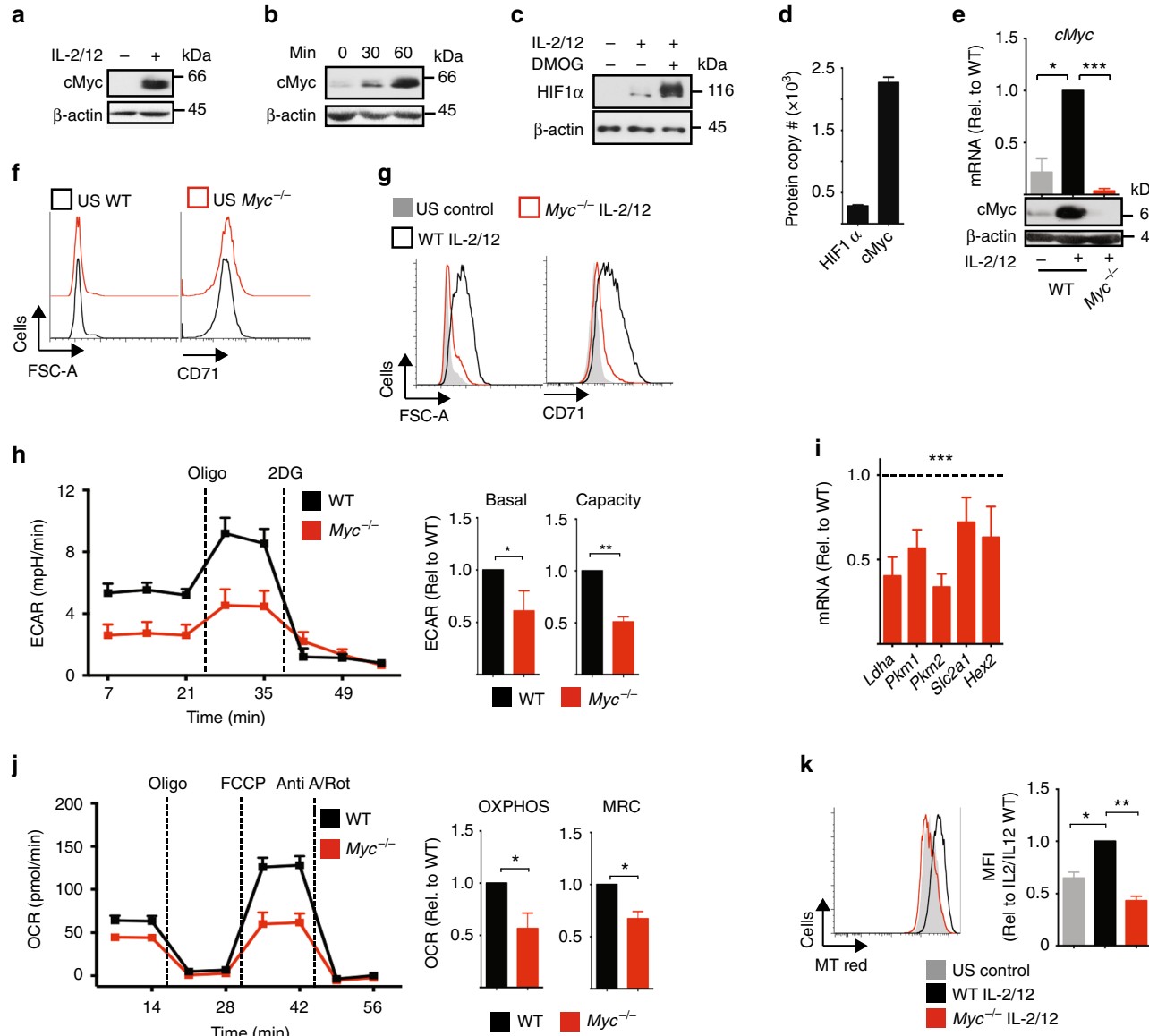

**Fig. 1** cMyc is required for IL-2/IL-12-induced NK cell metabolism. **a–c** NK cells were left unstimulated (US) or were stimulated with IL-2/IL-12 for 18 h (**a**, **c**) or for the time points indicated (**b**). DMOG (200 μM) was added for the last 2 h of activation as indicated (**c**). Samples were analysed by immunoblot for cMyc, HIF1α and β-actin protein expression. **d** NK cells were stimulated with IL-2/IL-12 for 18 h and protein copy numbers of HIF1α and cMyc per cell were determined using quantitative proteomic analysis. **e–k** $cMyc^{-/-}$ ($cMyc^{flox/flox}$ x Tamox-Cre) or WT ($cMyc^{WT/WT}$ x Tamox-Cre) NK cells were left unstimulated or stimulated for 18 h with IL-2/12. **e** qPCR and immunoblot analysis were used to determine the expression of $cMyc$ mRNA and cMyc protein, respectively. **f**, **g** Flow cytometry analysis of FSC-A and CD71 expression. **h** Analysis of NK cell extracellular acidification rate (ECAR) to assess basal glycolytic rate and glycolytic capacity. **i** qPCR analysis of mRNA for lactate dehydrogenase ($Ldha$), pyruvate kinases ($Pkm1$, $Pkm2$), the Glut1 glucose transporter ($Slc2a1$) and hexokinase 2 ($Hex2$). **j** Analysis of NK cell oxygen consumption rate (OCR) to assess rates of OXPHOS and maximal respiration. **k** Flow cytometry analysis of mitochondrial mass using MitoTracker Red (MT Red) staining. Data are mean ± s.e.m. of 3 (**d–k**) or 4 (**h**, **j**) experiments, or representative of 3 (**a–c**, **k**), 4 (**h**, **j**) or 6 (**f**, **g**) individual experiments. **h**, **j** For representative plots, the data were normalised to 200,000 cells. Statistical analysis was performed using a two-way ANOVA with Tukey post test (**e–k**) or one-sample $t$-test vs. a theoretical value of 1 (**h**, **j**); *$p < 0.05$, **$p < 0.01$, ***$p < 0.001$. Oligo oligomycin, 2DG 2-deoxyglucose, Anti A antimycin A, Rot rotenone, FCCP carbonyl cyanide-4-(trifluoromethoxy) phenylhydrazone

rate compared to WT controls (Fig. 1j, Supplementary Fig. 1d). The decreased mitochondrial function in $Myc^{-/-}$ NK cells was associated with decreased mitochondrial mass; IL-2/IL-12 stimulation resulted in an increase in mitochondrial mass, as measured using a mitochondrial specific dye, in WT NK cells but not in $Myc^{-/-}$ NK cells (Fig. 1k). Consistent with the observation that IL-2/IL-12-stimulated NK cells did not induce high levels of HIF1α expression, HIF1α-null NK cells ($Hif1a^{-/-}$) had no defect in IL-2/IL-12-induced metabolic responses (Supplementary Fig. 2a-g).

Previous work has shown that NK cell metabolism is integrally linked to NK cell effector function[1–3]. Therefore, we next considered whether $Myc^{-/-}$ NK cells had defective effector functions following IL-2/IL-12 cytokine stimulation. In the absence of cMyc, IL-2/IL-12-activated NK cells produced less IFNγ (Fig. 2a, b), and had reduced expression of the cytolytic molecule granzyme B compared to WT NK cells (Fig. 2c, d). In contrast, and consistent with the metabolic analysis of $Hif1a^{-/-}$ NK cells, there were no functional differences between WT and $Hif1a^{-/-}$ NK cells (Fig. 2e–h).

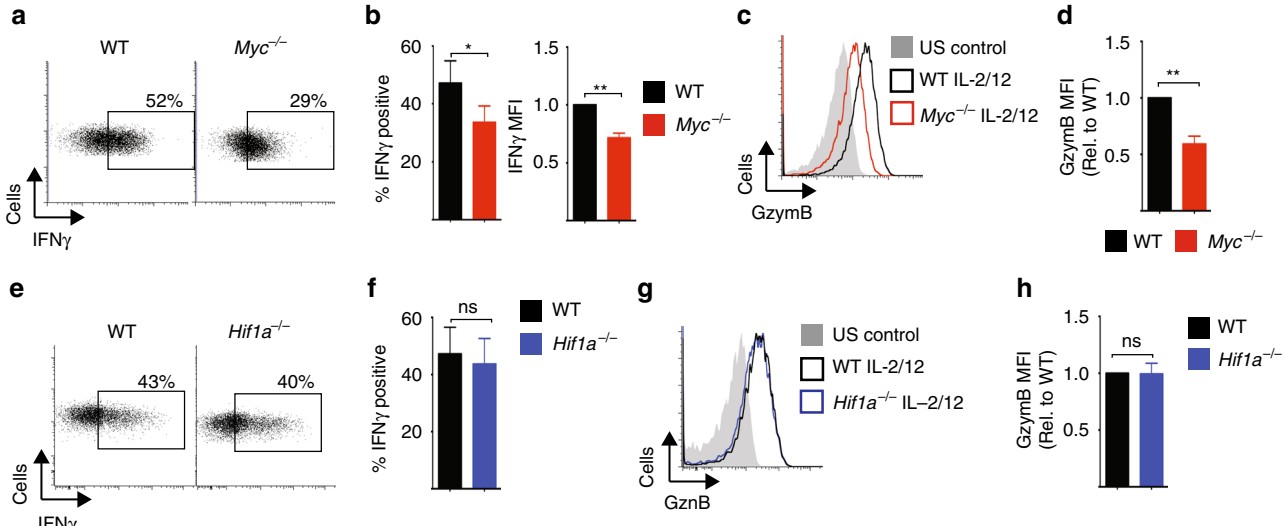

**Fig. 2** cMyc, but not HIF1α, is required for NK cell effector functions. **a–d** cMyc⁻/⁻ (cMyc^flox/flox × Tamox-Cre) or WT (cMyc^WT/WT × Tamox-Cre) NK cells were left unstimulated (US) or were stimulated for 18 h with IL-2/IL-12 before flow cytometry analysis for % IFNγ-positive NK cells and mean fluorescence intensity (MFI) of IFNγ expression in IFNγ⁺ NK cells, and granzyme B expression. **e–h** Hif1α⁻/⁻ (Hif1α^flox/flox × Vav-Cre) or WT (Hif1α^flox/flox) NK cells were left unstimulated or were stimulated for 18 h with IL-2/IL-12 before flow cytometry analysis for percent IFNγ-positive NK cells and granzyme B expression. Data are mean ± s.e.m. or representative of five independent experiments. Statistical analysis was performed using Student's t-test (**b**, **f**) or a one-sample t-test vs. a theoretical value of 1 (**b–h**); *p < 0.05, **p < 0.01, ns non-significant

Taken together, these data show that the transcription factor cMyc, but not HIF1α, is crucial for the IL-2/IL-12-dependent metabolic reprogramming of NK cells that accompanies robust NK cell functional responses.

**mTORC1 but not Akt signalling regulates cMyc expression.** We next considered the signalling mechanisms responsible for the robust cMyc expression in IL-2/IL-12-stimulated NK cells. In other cell types phosphatidylinositol-3-kinase (PI3K)/Akt and mTORC1 signalling have been linked to cellular metabolism through the regulation of cMyc expression[22,23]. To investigate this in NK cells, highly specific inhibitors of Akt or mTORC1 were used, Akti-1/2 and rapamycin, respectively. NK cells stimulated with IL-2/IL-12 for 18 h were then treated for 1 h with Akti-1/2 before immunoblot analysis. While effective Akt inhibition was confirmed by the loss of Akt phosphorylation on serine 473, no decrease in cMyc protein levels was observed (Fig. 3a). Similarly, NK cells stimulated for for 18 h with Akti-1/2 showed no decrease in cMyc levels (Fig. 3b). Consistent with normal cMyc expression, NK cells stimulated in the presence of Akti-1/2 were equivalent in size and CD71 expression, and produced comparable levels of IFNγ and granzyme B compared to control untreated NK cells (Fig. 3c–g). Interestingly, mTORC1-signalling was not decreased in the presence of Akti-1/2, as measured by the phosphorylation of the mTORC1 substrate p70 S6 Kinase and the S6K substrate S6 ribosomal protein (Fig. 3h). Therefore, we next considered whether mTORC1 signalling was important for cMyc expression in NK cells. Rapamycin-inhibited cMyc protein expression in NK cells stimulated with IL-2/IL-12 for 30 min and 1 h (Fig. 3i). Therefore, mTORC1 activity is required for the initial IL-2/IL-12-stimulated upregulation of cMyc. Next, we determined whether cMyc protein expression remained dependent on mTORC1 activity over longer time periods. Interestingly, after 18 h of IL-2/IL-12 stimulation in the presence of rapamycin, NK cells now expressed high levels of cMyc, indicating that mTORC1 activity is not required for sustained cMyc expression in IL-2/IL-12-stimulated NK cells (Fig. 3j). Therefore, the data indicate that mTORC1 is important

for initial IL-2/IL-12-induced cMyc but that additional mechanisms are involved in promoting cMyc protein expression after prolonged stimulation.

**Amino acid transport through SLC7A5 controls cMyc levels.** cMyc protein expression has been reported to be sensitive to levels of amino acids in T lymphocytes. In particular, transport through the system L-amino acid transporter SLC7A5 is essential for maintaining cMyc levels in IL-2-maintained CD8+ CTLs[24]. Therefore, we considered whether amino acid availability is important for cMyc protein expression in NK cells. The hetero-dimeric system L-amino acid transporters are composed of a common heavy chain subunit SLC3A2 (CD98) and a light chain subunit responsible for amino acid transport: SLC7A5 (LAT1), SLC7A8 (LAT2), SLC43A1 (LAT3) and SLC43A2 (LAT4). Our proteomics data show that NK cells only express SLC7A5 (LAT1) (Fig. 4a). This correlates with mRNA expression of the system L transporters available in the Immgen database. Slc7a5 mRNA expression is robustly increased after 18 h of IL-2/IL-12 stimulation and continued IL-2 signalling is required to maintain SLC7A5 expression (Fig. 4b, c). This induction of SLC7A5 expression was confirmed by measuring the transport capacity of the system L substrate 3H-phenylalanine, which was also robustly induced in IL-2/IL-12-stimulated NK cells (Fig. 4d). 2-Amino-2-norbornanecarboxylic acid (BCH), a competitive blocker of system L-amino acid transport, was used as a negative control and prevented 3H-phenylalanine uptake (Fig. 4d). BCH can be considered a specific inhibitor of uptake through the SLC7A5/SLC3A2 (LAT1) transporter in NK cells, which do not express LAT2–4 (Fig. 4a). Next, BCH was used to determine the importance of SLC7A5-mediated amino acid transport for maintaining cMyc expression. When cytokine-activated NK cells were treated with BCH a dramatic decrease in cMyc protein levels was observed after just 30 min (Fig. 4e). BCH treatment also resulted in reduced mTORC1 signalling (Fig. 4e). Additionally, SLC7A5-null (from Slc7a5^flox/flox Vav-Cre mice) NK cells stimulated for 18 h with IL-2/IL-12 did not induce cMyc protein expression (Fig. 4f). One reason why SLC7A5 is required for

mTORC1 activity in CD8+ T cells is that it transports the amino acid leucine into the cell and there are well-described mechanisms linking leucine to the regulation of mTORC1 activity[25]. Indeed,

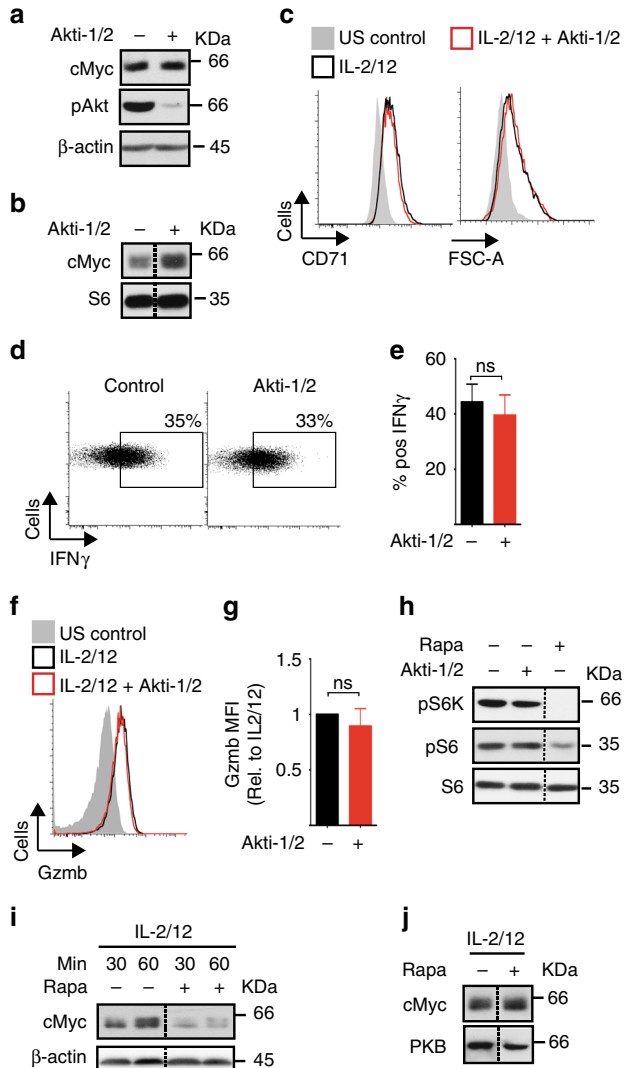

**Fig. 3** mTORC1, but not Akt, is required for cMyc expression in IL-2/IL-12-stimulated NK cells. **a** NK cells were left unstimulated (US) or stimulated with IL-2/IL-12 for 18 h and Akti-1/2 (2 μM) was added for the last 1 h of activation, before immunoblot analysis for levels of cMyc, phosphorylated Akt on serine 473 (pAkt) and β-actin. **b** NK cells were stimulated with IL-2/IL-12 in the presence or absence of Akti-1/2 (2 μM) for 18 h before immunoblot analysis of cMyc and S6 ribosomal protein (S6, loading control) levels. **c–g** NK cells were left unstimulated or stimulated with IL-2/IL-12 in the presence or absence of Akti-1/2 (2 μM) for 18 h, then analysed by flow cytometry for FSC-A and CD71 expression (**c**), IFNγ production (**d**, **e**) and granzyme B expression (**f**, **g**). **h** NK cells were left unstimulated or stimulated with IL-2/IL-12 for 18 h and Akti-1/2 (2 μM) or rapamycin (20 nM) were added for the last hour of activation as indicated. Samples were subjected to immunoblot analysis for levels of phosphorylated S6 ribosomal protein on serine 235/6 (pS6), phosphorylated p70 S6 kinase on serine 389 (pS6K) and S6 ribosomal protein (S6). **i**, **j** NK cells were stimulated with IL-2/IL-12 for 30 min, 60 min (**i**) or 18 h (**j**) in the presence or absence of rapamycin (20 nM) as indicated before immunoblot analysis for cMyc and β-actin expression. Data are mean ± s.e.m of 6 experiments (**e**, **g**), or representative or 3 (**a–j**) or 6 (**c–f**) individual experiments. Statistical analysis was performed using Student's t-test (**e**) or a one-sample t-test vs. a theoretical value of 1 (**g**); ns non-significant

NK cells activated in media without leucine were deficient for mTORC1 signalling but, interestingly, these cells had normal levels of cMyc (Fig. 4g). SLC7A5 is an obligate anti-porter, which means that it must transport an amino acid out of the cell, most commonly glutamine, in order to transport another amino acid into the cell[26]. Therefore, intracellular glutamine is important in sustaining SLC7A5-mediated amino acid transport. IL-2/IL-12 stimulation robustly increased the rate of glutamine transport into the cell (Fig. 4h). Glutamine uptake was not mediated by system L transporters as it was not affected by BCH treatment (Fig. 4h). Glutamine is predominantly transported by system ASC (SLC1A4 and SLC1A5) and system A and N (SLC38A1, SLC38A2, SLC38A5 and SLC38A7) transporters[27,28]. Quantitative proteomics data showed that the predominant glutamine transporters in activated NK cells are SLC1A5 and SLC38A2 (Supplementary Fig. 3a). Next, we tested whether glutamine is important for sustaining cMyc protein expression in IL-2/IL-12-activated NK cells. Glutamine withdrawal resulted in a rapid decrease in cMyc protein expression within 30 min (Fig. 4i). In contrast, when IL-2/IL-12-activated NK cells were deprived of leucine or treated with rapamycin for 1 h, cMyc protein expression was not decreased (Fig. 4j). Together, these data show that glutamine, but not mTORC1 activity, is required for sustaining cMyc expression in cytokine active NK cells (Figs. 3 and 4). In T cells, glutamine has been shown to regulate cMyc expression through a mechanism involving fuelling of the hexosamine pathway to support protein O-GlcNAcylation[29]. To investigate whether the loss of cMyc protein following glutamine withdrawal might be explained by decreased protein O-GlcNAcylation, a flow cytometry-based assay was used to measure global O-GlcNAcylation levels. While glutamine withdrawal does result in reduced protein O-GlcNAcylation after 18 h, shorter periods of glutamine withdrawal did not affect protein O-GlcNAcylation, indicating that this mechanism is unlikely to account for the rapid loss in cMyc protein levels (Fig. 4i, j, Supplementary Fig. 3b). Taken together, the data argue that while SLC7A5 transport of leucine is required for mTORC1 signalling, glutamine and the transport of other amino acids through SLC7A5 are required to sustain cMyc protein expression.

Given that cMyc is crucial for metabolic reprogramming in IL-2/IL-12-stimulated NK cells, we next considered the impact of BCH on NK cell metabolic responses. NK cells stimulated with IL-2/IL-12 in the presence of BCH for 18 h were reduced in size and had decreased expression of CD71 compared to untreated controls (Fig. 5a). Inhibition of system L-amino acid uptake also resulted in reduced levels of cellular metabolism; BCH treated NK cell had reduced rates of glycolysis and glycolytic capacity, and reduced rates of OXPHOS and maximal respiration (Fig. 5b–e). These data are consistent with our observations in Myc[−/−] NK cells (Fig. 1). Similar results were obtained when NK cells were stimulated with IL-2/IL-12 in the absence of glutamine (Supplementary Fig. 3c–g). Consistent with these observed metabolic defects, NK cells stimulated with IL-2/IL-12 in the presence of BCH or the absence of glutamine had reduced IFNγ production and granzyme B expression (Fig. 5f–i Supplementary Fig. 3h, i).

Next we performed experiments to confirm that amino acid-controlled cMyc is also required for metabolic and functional NK cell responses in NK cells isolated directly from mouse spleens. Splenocytes from green fluorescent protein (GFP)-cMyc reporter mice were used to measure cMyc levels in NK cells. cMyc protein expression was significantly increased in splenic NK cells stimulated with IL-2/IL-12 for 18 h, but not in those stimulated in the presence of BCH (Fig. 6a, b). Additionally, the expression of the cMyc target gene CD71 was found to parallel the observed cMyc protein expression (Fig. 6c). The metabolic pathways of

splenic NK cells stimulated with IL-2/IL-12 for 18 h were then analysed. As seen with cultured NK cells, IL-2/IL-12-stimulated splenic NK cells showed robust increases in the rates of both glycolysis and OXPHOS as well as in glycolytic capacity and maximal respiration rates (Fig. 6d–g). This metabolic response was prevented when system L-amino acid transport was inhibited with BCH in line with the fact that these NK cells do not express cMyc (Fig. 6d–g). Parallel experiments showed that BCH treatment also inhibited IFNγ production and granzyme B expression in IL-2/IL-12-stimulated splenic NK cells (Fig. 6h, i).

**cMyc protein is controlled by GSK3-targeted degradation**. The data show that in activated NK cells, cMyc protein levels are rapidly lost following BCH treatment or glutamine withdrawal arguing that cMyc is being actively degraded. Indeed, studies in

other cell types suggest that cMyc protein levels are extremely labile and determined by constitutive cMyc protein synthesis and degradation[30]. To investigate whether proteasomal degradation plays a role in the regulation of cMyc levels, IL-2/IL-12-activated NK cells were treated with BCH or deprived of glutamine in the presence or absence of the proteasome inhibitor MG132. In the control cells, MG132 treatment resulted in a significant increase in cMyc, confirming that cMyc is actively degraded in these NK cells (Supplementary Fig. 4a, b). Additionally, in NK cells treated with BCH or deprived of glutamine for 1 h, MG132 treatment resulted in a significant rescue of cMyc protein levels (Supplementary Fig. 4a, b). The activity of glycogen synthase kinase 3 (GSK3) has been linked to the regulation of cMyc degradation. GSK3 has been reported to phosphorylate cMyc on serine 58 to promote degradation in the proteasome[31,32]. Indeed, the highly specific GSK3 inhibitor CT99021[33] prevented the rapid decrease in cMyc protein expression in cells treated with BCH or deprived of glutamine (Supplementary Fig. 4c, d). Together, these data argue that a balance between high rates of cMyc synthesis and GSK3-targeted cMyc degradation determines cMyc protein levels in activated NK cells.

**Glutaminolysis does not sustain OXPHOS in activated NK cells**. These data show a role for glutamine in controlling cMyc-dependent metabolic responses in IL-2/IL-12-stimulated NK cells. As glutamine is also a fuel that feeds into the TCA cycle through glutaminolysis in other lymphocyte subsets[14,34], we next considered the relative importance of glutamine in NK cells as a signalling molecule to sustain cMyc protein expression vs. a fuel to sustain OXPHOS. NK cells stimulated with IL-2/IL-12 for 18 h were switched into glutamine-free media for 1 h prior to metabolic flux analysis for rates of OXPHOS. Alternatively, NK cells

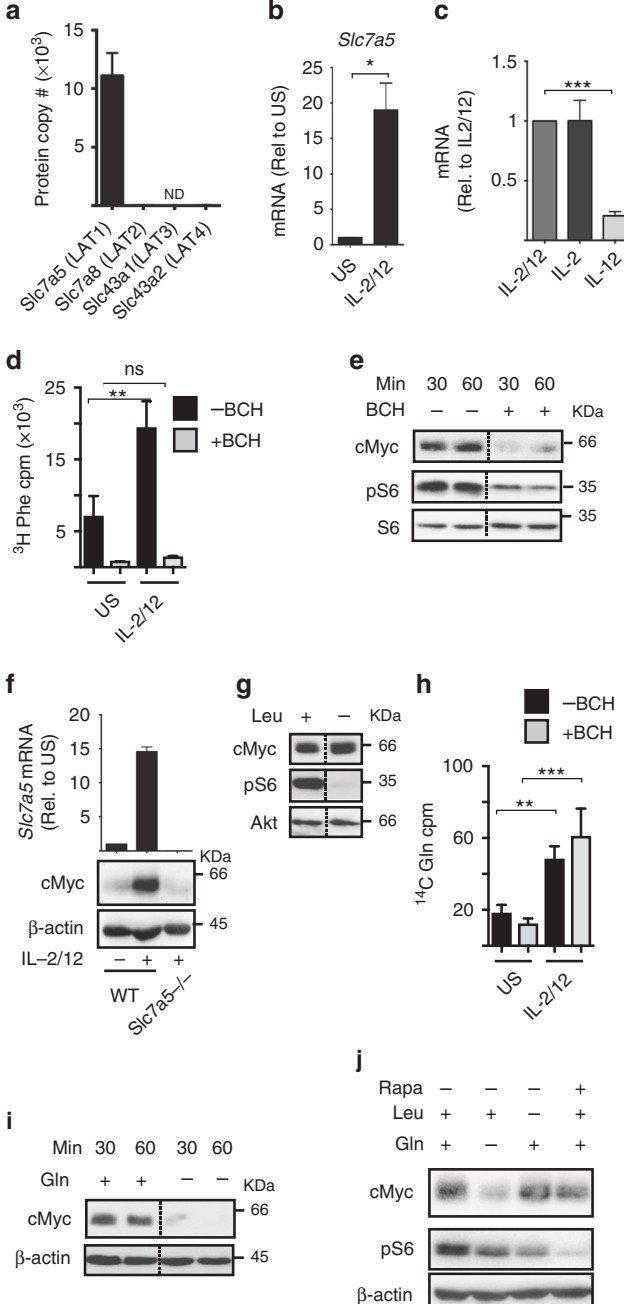

**Fig. 4** SLC7A5 activity is required for IL-2/IL-12-induced cMyc expression. **a** NK cells were activated for 18 h with IL-2/IL-12 and the number of protein copies per cell were determined using quantitative proteomic analysis: SLC7A5, SLC7A8, SLC43A1 and SLC43A2. **b** NK cells were left unstimulated (US) or were stimulated with IL-2/IL-12 for 18 h and *Slc7a5* mRNA levels analysed by qPCR. **c** NK cells were activated with IL-2/IL-12 for 20 h and then switched into media containing cytokines as shown for a further 8 h. *Slc7a5* mRNA levels were analysed by qPCR. **d** Purified NK cells were stimulated with IL-2/IL-12 for 18 h and uptake of ³H-labelled phenylalanine was measured in the presence or absence of the system L competitor BCH (10 mM). **e** NK cells were stimulated with IL-2/IL-12 for 18 h and the system L blocker BCH (25 mM) was added for the final 30 or 60 min as indicated before immunoblot analysis of cMyc, phosphorylated S6 ribosomal protein on serine 235/6 (pS6) and total S6 ribosomal protein (S6). **f** *Slc7a5⁻/⁻* (*Slc7a5^flox/flox* × Vav-Cre) or WT (*Slc7a5^WT/WT* × Vav-Cre) NK cells were left unstimulated or were stimulated for 18 h with IL-2/12 before immunoblot analysis of cMyc and β-actin protein expression and qPCR analysis of *Slc7a5* mRNA expression. **g** NK cells were stimulated with IL-2/IL-12 in the presence or absence of leucine for 18 h before immunoblot analysis of cMyc, pS6 and Akt. **h** Purified NK cells were stimulated with IL-2/IL-12 for 18 h and glutamine uptake was measured using ¹⁴C-labelled glutamine in the presence or absence of BCH (10 mM). **i** NK cells stimulated with IL-2/IL-12 for 18 h were cultured in the presence or absence of glutamine for 30 or 60 min as indicated, before immunoblot analysis for levels of cMyc and β-actin. **j** IL-2/IL-12-activated NK cells were cultured for 1 h in IL-2/IL-12 media with or without the amino acids L-glutamine or L-leucine or the addition of rapamycin. Data are mean ± s.e.m of 3 experiments (**a**–**h**) or is representative of 2 (**f**) or 3 individual experiments (**e**–**j**). Statistical analysis was performed using a one-way ANOVA with Tukey post test (**b**–**h**) or Student's *t*-test (**a**); *$p < 0.05$, **$p < 0.01$, ***$p < 0.005$, ns non-significant, ND not detected

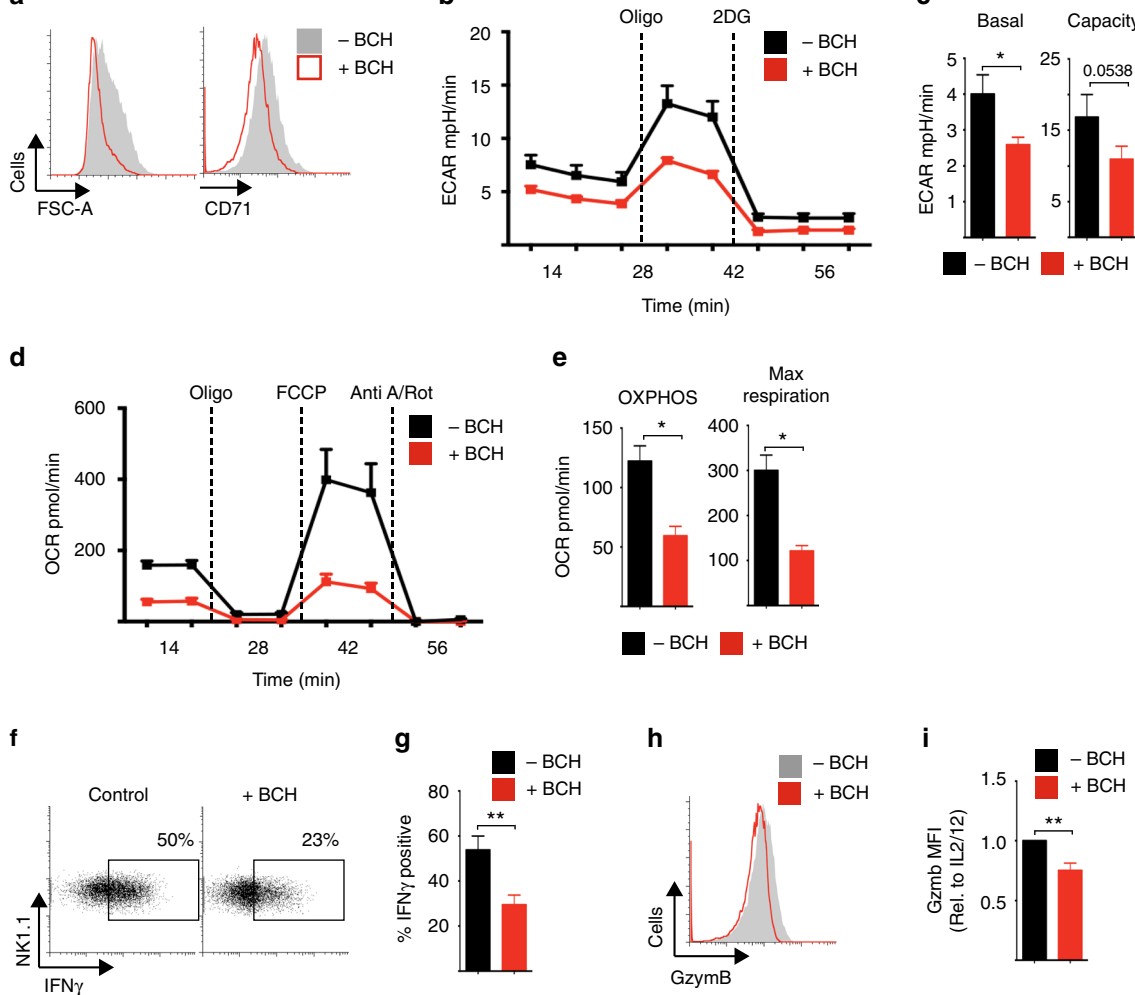

**Fig. 5** SLC7A5 is required for NK cell metabolic and functional responses. **a–i** NK cells were stimulated with IL-2/IL-12 in the presence or absence of BCH (25 mM) for 18 h as indicated. **a** FSC-A and CD71 expression were analysed by flow cytometry. **b, c** Analysis of extracellular acidification rate (ECAR) of NK cells to assess basal glycolytic rate and glycolytic capacity. **d, e** Analysis of NK cell oxygen consumption rate (OCR) to assess rates of OXPHOS and maximal respiration. **f, g** IFNγ and **h, i** granzyme B expression were analysed by flow cytometry. Data are mean ± s.e.m. of 4 (**c, e**) or 5 (**g, i**) experiments or representative of 4 (**b, d**) or 5 (**a–h**) individual experiments. **b–e** Data were normalised to 200,000 cells. Statistical analysis was performed using Student's *t*-test (**c–g**) or a one-sample *t*-test vs. a theoretical value of 1 (**i**); *$p < 0.05$, **$p < 0.01$. Oligo oligomycin, 2DG 2-deoxyglucose, Anti A antimycin A, Rot rotenone, FCCP carbonyl cyanide-4-(trifluoromethoxy)phenylhydrazone

were treated with the glutaminase inhibitor bis-2-(5-phenylacetamido-1,3,4-thiadiazol-2-yl)ethyl sulphide (BPTES), which inhibits the first enzyme of the glutaminolysis pathway, or a glutamine analogue that is a general inhibitor of glutamine-utilising enzymes, 6-diazo-5-oxo-L-norleucine (DON). The rate of OXPHOS was not significantly affected by glutamine deprivation or the addition of BPTES or DON (Fig. 7a–d, Supplementary Fig. 5a, b). The inhibition of glutaminase with BPTES, but not glutamine deprivation or DON treatment, had a small but significant impact upon the maximal respiration rate (Fig. 7a–d, Supplementary Fig. 5b). Glutamine is an important fuel in T cells and, indeed, BPTES and DON treatment inhibited T-cell proliferation (Supplementary Fig. 5c, d). These data suggested that glutamine is not a key fuel for sustaining elevated levels of OXPHOS in cytokine-stimulated NK cells.

To further explore glutamine metabolism we performed a $^{13}$C carbon-tracing metabolomics experiment using uniformly labelled $^{13}$C$_5$-glutamine. NK cells were stimulated for 16 h with IL-2/IL-12 in normal media and then switched into media containing $^{13}$C$_5$-glutamine for 30 min, 2 h or 8 h. The data

showed that glutamine does feed into the TCA cycle in NK cells giving rise to m+4 labelled TCA cycle intermediates (Fig. 7e). However, the mean enrichment of $^{13}$C within citrate was significantly less than in other TCA cycle intermediates (Fig. 7f). These data fit with the recent report that demonstrated that IL-2/IL-12-stimulated NK cells metabolise glucose to generate cytosolic citrate through the citrate malate shuttle[2]. Indeed, when comparing the incorporation of $^{13}$C from $^{13}$C$_6$-glucose and $^{13}$C$_5$-glutamine into the TCA cycle it is apparent that while the carbons from glutamine feed into the TCA cycle, the carbons from glucose are enriched in citrate and are not passed to downstream TCA cycle intermediates (Fig. 7g)[2]. Taken together, these data argue that cytokine-activated NK cells concurrently utilise the glucose-fuelled citrate malate shuttle and the glutamine-fuelled TCA cycle (Fig. 7e–g)[2].

Considering that the citrate malate shuttle can also drive OXPHOS[2], we next explored the relative contributions of the glutamine-fuelled TCA cycle and the glucose-fuelled citrate malate shuttle in sustaining elevated OXPHOS in IL-2/IL-12-activated NK cells. NK cells were stimulated with IL-2/IL-12 for

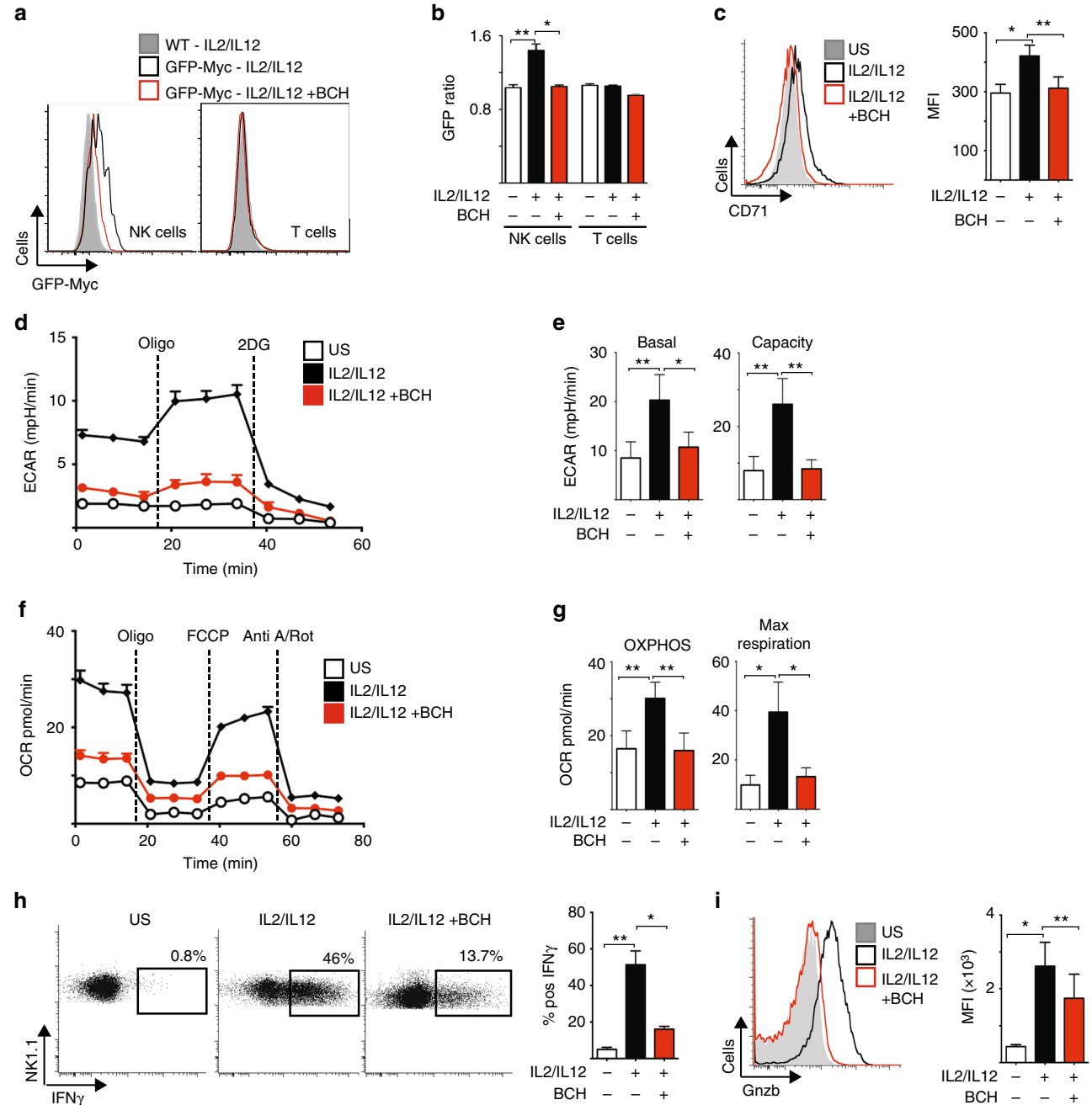

**Fig. 6** cMyc is required for splenic NK cell metabolic and functional responses. **a**, **b** Ex vivo splenic NK cells and T cells were isolated from CD45.2 GFP-Myc reporter mice or CD45.1 WT mice, mixed in a 1:1 ratio and activated for 18 h with IL-2/12 in the presence or absence of BCH. The ratio of fluorescence in CD45.2 and CD45.1 cells was calculated to give a measure of cMyc expression (CD45.2 GFP-Myc) adjusted for autofluorescence (CD45.1 WT). **c–i** Ex vivo NK cells were left unstimulated (US) or were activated for 18 h with IL-2/12 in the presence or absence of BCH. **c** Flow cytometry was used to measure the expression of CD71. **d**, **e** Analysis of NK cell extracellular acidification rate (ECAR) to assess basal glycolytic rate and glycolytic capacity. **f**, **g** Analysis of NK cell oxygen consumption rate (OCR) to assess rates of OXPHOS and maximal respiration. Flow cytometry was used to measure the expression of IFNγ (**h**) and granzyme B (**i**). Data are mean ± s.e.m. of 5 (**b**) or 6 (**c, e, g-i**) or representative 5 (**a**) or 6 (**d, f, h, i**) mice. **d–g** Data were normalised to 200,000 cells. Statistical analysis was performed using a one-way ANOVA with Tukey post test (**b, c, e, g, h, i**); *$p < 0.05$, **$p < 0.01$. Oligo oligomycin, 2DG 2-deoxyglucose, Anti A, antimycin A, Rot rotenone, FCCP carbonyl cyanide-4-(trifluoromethoxy)phenylhydrazone

18 h and then metabolic rates were measured before and after the addition of BPTES to inhibit glutamine-fuelled TCA cycle, or SB204990 to inhibit ACLY and the citrate malate shuttle. The addition of SB204990, but not BPTES, resulted in a decrease in basal OXPHOS rates (Fig. 7h). In contrast to the small effect of BPTES, SB204990 treatment resulted in a substantial decrease in the maximal respiration rate (Fig. 7h). Another ACLY inhibitor BMS303141 resulted in a similarly large decrease in maximal

respiration rates (Supplementary Fig. 5e). Together, these data indicate that OXPHOS is primarily maintained by the glucose-fuelled citrate malate shuttle, and not glutaminolysis, in IL-2/IL-12-activated NK cells. If so, it would be predicted that the concentrations of citrate and malate would be substantially greater than those of other TCA cycle metabolites due to the fact that they are also part of the citrate malate shuttle (Supplementary Fig. 5f). Indeed, this was found to be the case; citrate and

malate were present at 5 times the concentration of succinate and fumarate (Fig. 7i).

**Glutamine-dependent cMyc sustains NK cell responses.** While these data argue that glutamine is not an important fuel in NK cells, glutamine is required for the expression of the metabolic regulator cMyc (Fig. 1). Therefore, we investigated whether glutamine is required for maintaining prolonged NK cell metabolism through the regulation of cMyc. NK cells were stimulated with IL-2/IL-12 for 20 h and then cultured for a further 20 h in glutamine-

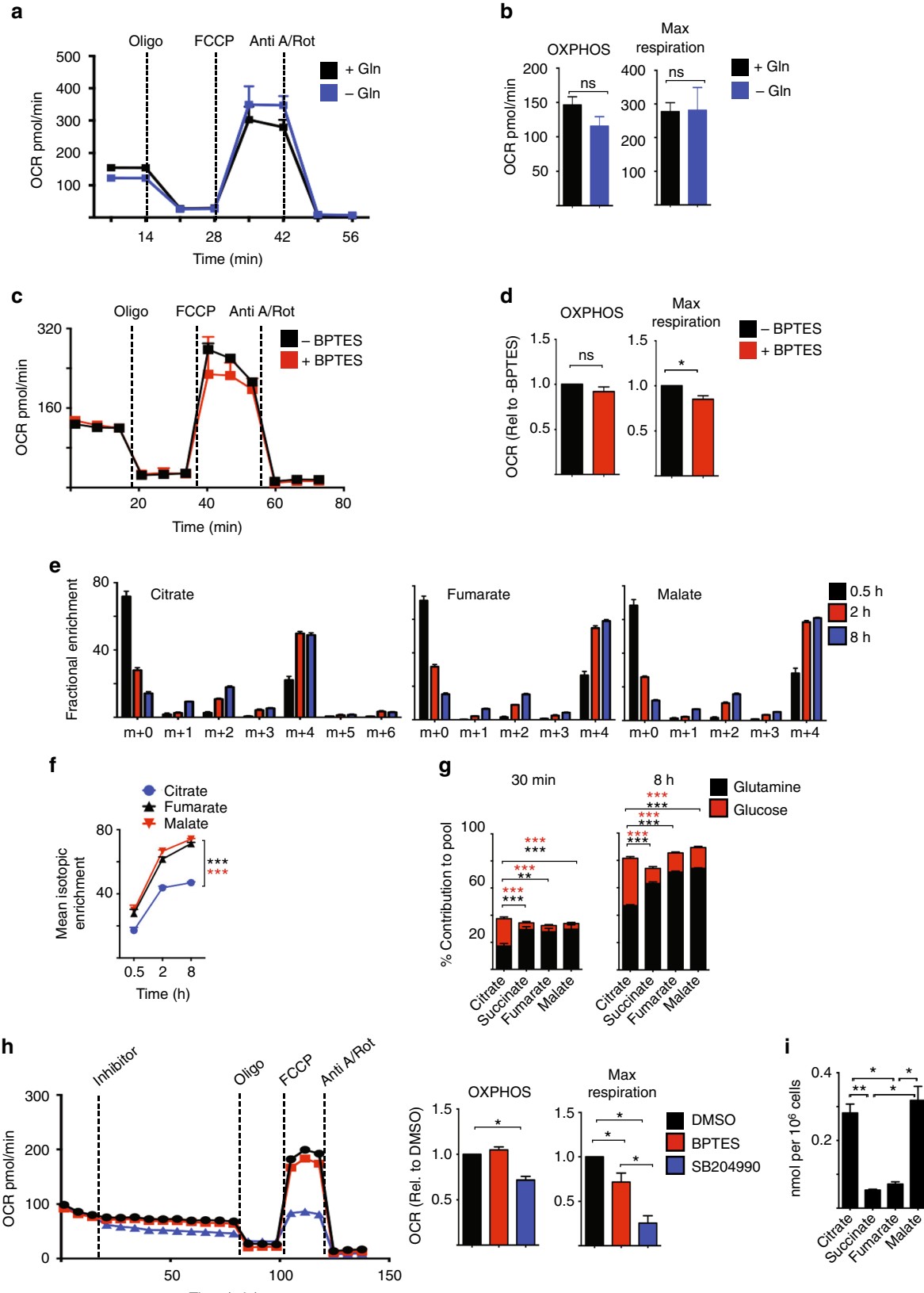

free media or in media with glutamine in the presence or absence of BPTES or DON. NK cells deprived of glutamine were small compared to NK cells cultured with glutamine (Fig. 8a). In contrast, NK cells cultured with glutamine and treated with BPTES or DON were comparable in size to control cells (Fig. 8a, Supplementary Fig. 6a). IL-2/IL-12-activated NK cells treated with BPTES or DON for 1 h maintain high levels of cMyc protein expression (Fig. 8b, Supplementary Fig. 6b) unlike those deprived of glutamine for 1 h (Fig. 4i). Additionally, the metabolic signature of IL-2/IL-12-activated NK cells deprived of glutamine for 20 h was very different from those treated with BPTES for 20 h. Consistent with the data showing that glutaminolysis does feed into the TCA cycle and has a minor role in supporting OXPHOS, BPTES treatment for 20 h resulted in a small decrease in both OXPHOS and maximal respiration rates (Fig. 8c). However, the impact of glutamine deprivation was much greater than that of BPTES treatment; NK cells deprived of glutamine had a twofold decrease in OXPHOS and a threefold decrease in maximal respiration (Fig. 8c). The data presented herein suggest that the substantial effect of glutamine deprivation on NK cell metabolism is due to the loss of cMyc expression in glutamine-deprived NK cells. Supporting this, glutamine-deprived NK cells but not BPTES-treated NK cells had decreased rates of glycolysis and glycolytic capacity (Fig. 8d), which are controlled by cMyc in NK cells (Fig. 1g, h). IL-2/IL-12-activated NK cells that were deprived of glutamine for 20 h had reduced levels of IFNγ production and expressed lower levels of granzyme B (Fig. 8e, f). In contrast, IL-2/IL-12-activated NK cells treated with BPTES for 20 h maintained these NK cell effector functions (Fig. 8e, f). Additionally, glutamine-deprived but not BPTES-treated NK cells had reduced cytotoxicity towards tumour target cells (Fig. 8g). Similar results were obtained using the more general inhibitor of glutamine metabolism, DON (Supplementary Fig. 8c–g).

Taken together, these data show that glutamine availability, but not glutamine metabolism, is crucial for NK cell anti-tumour functions.

## Discussion

In this study, we have shown a key role for cMyc in supporting the metabolic changes required for NK cell responses following IL-2/IL-12 cytokine stimulation (Supplementary Fig. 7). This combination of cytokines drives a robust metabolic response because IL-12 induces the expression of the high-affinity IL-2 receptor CD25 facilitating IL-2-dependent metabolic reprogramming[1]. Interestingly, the high rates of cellular glycolysis in IL-2-maintained CTLs are dependent on HIF1α rather than cMyc[12]. Nevertheless, cMyc is important for metabolic responses in other lymphocyte subsets such as in TCR-stimulated T cells[14]. It is worth noting that while the IL-2 stimulus is shared between IL-2/IL-12-stimulated NK cells and IL-2-maintained CTLs, the metabolic requirements of these cells will be distinct, as NK cells are engaging in blastogenesis while CTLs are rapidly proliferating. Both cMyc and HIF1α promote glycolysis and have overlapping target genes but they also have unique functions. For instance, cMyc, and not HIF1α, has been linked to the regulation of mitochondrial biogenesis and OXPHOS[35]. Therefore, it is likely that cMyc is essential at this early point during NK cell activation as it promotes a metabolic response that matches the metabolic requirements of cellular blastogenesis. While the data show that HIF1α is not required for the initial metabolic response accompanying NK cell activation, this does not preclude a role for HIF1α at other points during NK cell response. For instance, it would be of interest to investigate whether HIF1α plays a role in responses mediated by trained (or memory-like) NK cells.

cMyc protein expression can be regulated by both translational and multiple post-translational mechanisms[36]. The data show that mTORC1 activity is required for the initial increase in cMyc protein expression following NK cells activation with IL-2/IL-12. mTORC1 signalling regulates 5' cap-dependent translation, which is important for the translation of cMyc mRNA[37–39]. However, after prolonged periods of cytokine stimulation mTORC1 signalling is not required for cMyc protein expression. Our previous work showed that in CTLs, cMyc protein levels are also independent of mTORC1 signalling[12,40]. Interestingly, in transformed CD8+ leukaemic T cells, cMyc protein expression is dependent on mTORC1 activity[41]. This highlights that in CD8+ T cells, depending on the context, cMyc protein expression can also be regulated by mTORC1-dependent and -independent mechanisms. Herein, we show that in activated NK cells, cMyc protein is continuously subjected to GSK3-targeted proteasomal degradation. This means that to maintain high levels of cMyc protein, NK cells must sustain high rates of protein synthesis. The delivery of amino acids, which are essential for supporting high rates of protein synthesis, are required to sustain high levels of cMyc in NK cells. IL-2/IL-12-stimulated NK cells upregulate the expression and activity of the SLC7A5 amino acid transporter and SLC7A5-mediated amino acid transport facilitates elevated cMyc protein expression. In CTLs, SLC7A5-mediated amino acid transport has previously been shown to be critical for cMyc expression and for metabolic and functional responses[40]. Interestingly, SLC7A5 mRNA expression is also robustly induced in NK cells in mice following murine cytomegalovirus infection (www.immgen.org), suggesting that this transporter may have an important role in NK cell responses to viral infection.

This study shows that PI3K/Akt signalling is not required for IL-2/IL-12-induced NK cell metabolic or effector responses. This is in contrast to our previous research showing an important role for mTORC1 for cytokine-induced NK cell responses[1]. It is

---

**Fig. 7** Glutaminolysis is not important for sustaining OXPHOS in cytokine-activated NK cells. **a–d** NK cells were stimulated with IL-2/IL-12 for 18 h and then put in the presence or absence of glutamine (**a**, **b**) or in the presence or absence the glutaminase inhibitor BPTES (**c**, **d**) for 1 h prior to analysis of oxygen consumption rates (OCR) to assess rates of OXPHOS and maximal respiration. **e**, **f** Metabolic-tracing experiments were performed on NK cells stimulated with IL-2/IL-12 for 16 h in media with unlabelled glutamine and then further cultured for 0.5 h to 8 h in media containing $^{13}C_5$-glutamine. Analysis of the isotopologue distribution (**e**) of citrate, fumarate and malate was performed or isotopic mean enrichment of citrate, fumarate and malate (**f**) was calculated. **g** Analysis of $^{13}C$ incorporation into the TCA cycle intermediates citrate, succinate, fumarate and malate in 16 h IL-2/IL-12 activated NK cells cultured in $^{13}C_5$-glutamine or $^{13}C_6$-glucose for 30 min and 8 h. Data for the incorporation of $^{13}C_6$-glucose in the TCA cycle intermediates have been previously published[2] and are used here for direct comparison to the new data set of $^{13}C_5$-glutamine incorporation in TCA cycle intermediates. **h** NK cells were activated for 18 h with IL-2/IL-12 before the analysis of NK cell oxygen consumption rate (OCR) to assess rates of OXPHOS and maximal respiration following the injection of BPTES or SB204990. **i** NK cells stimulated with IL-2/IL-12 for 18 h were analysed by GC-MS for total concentrations of citrate, succinate, fumarate and malate. Data are mean ± s.e.m. of 4 (**a–i**), 5 (**d**) 6 (**e**, **f**) or 3–6 (**g**, **h**) experiments or representative of 3–6 (**c**, **h**) or 4 (**a**) individual experiments. **a–h** Data were normalised to 200,000 cells. Statistical analysis was performed using Student's $t$-test (**b**, **g**), a one-sample $t$-test (**d**, **h**), a one-way ANOVA with Tukey test (**g**, **i**) or a two-way ANOVA with Sidak test (**f**, **g**); *$p < 0.05$, **$p < 0.01$, ***$p < 0.005$, ns non-significant. Oligo oligomycin, 2DG 2-deoxyglucose, Anti A antimycin A, Rot rotenone, FCCP carbonyl cyanide-4-(trifluoromethoxy)phenylhydrazone

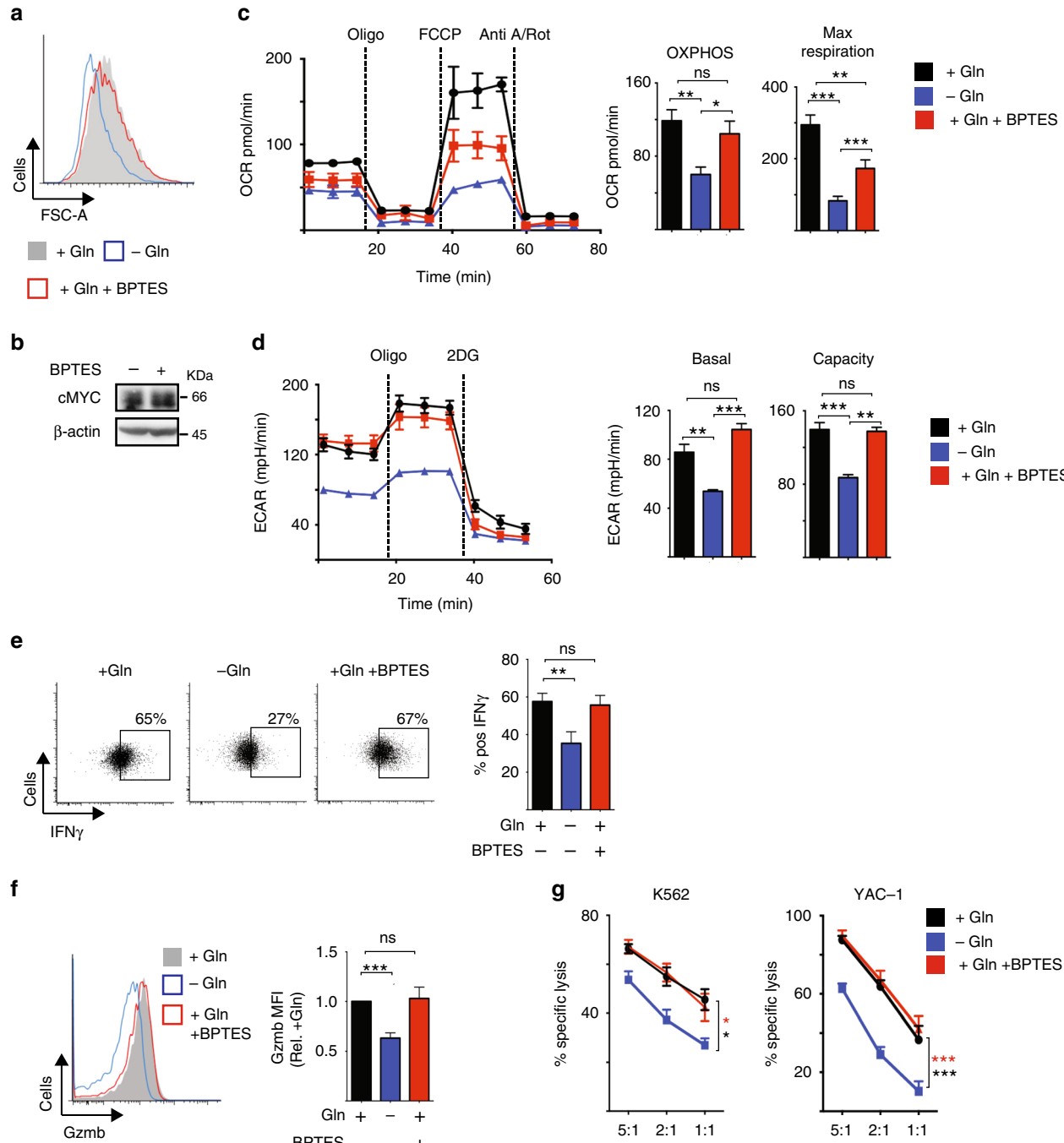

**Fig. 8** Glutamine-dependent cMyc sustains NK cell metabolism and effector function. NK cells were activated with IL-2/IL-12 for 20 h and then put in the presence or absence of glutamine with or without BPTES for a further 1 h (**b**) or 20 h (**a**, **c–g**). **a** Flow cytometry analysis of FSC-A. **b** Western blot analysis of cMyc and β-actin protein expression. **c** Analysis of NK cell oxygen consumption rate (OCR) to assess rates of OXPHOS and maximal respiration. **d** Analysis of NK cell extracellular acidification rates (ECAR) to assess basal glycolytic rate and glycolytic capacity. **e**, **f** Flow cytometry analysis of IFNγ production (**e**) and granzyme b expression (**f**). **g** The cytotoxicity of NK cells towards tumour cells was assessed; NK cells were activated as described and were incubated with either YAC-1 or K562 tumour cells at the ratios indicated and the percent-specific cell lysis calculated. **c**, **d** For representative plots the data were normalised to 200,000 cells. Data are mean ± s.e.m. of 5 (**g**), 7 (**c**, **d**) or 12 (**e**, **f**) experiments or representative of 6 (**b**), 7 (**c**, **d**) or 12 (**a–f**) individual experiments. Statistical analysis was performed using a one-sample *t*-test (**f**), a one-way ANOVA with Tukey test (**c**, **d**)) or Sidak test (**e**), or a two-way ANOVA with Tukey test (**g**); *$p < 0.05$, **$p < 0.01$, ***$p < 0.005$, ns non-significant

commonly believed that the mTORC1 kinase is activated downstream of PI3K/Akt signalling in immune cells, but this is not always the case. This idea of a PI3K/Akt/mTORC1 linear pathway is perpetuated by the fact that mTORC1 and PI3K/Akt signalling can have overlapping functions in lymphocytes, though the mechanisms involved can be quite distinct[12,42–44]. Here, we show

that mTORC1 signalling is robustly induced in IL-2/IL-12-stimulated NK cells even in the presence of an Akt inhibitor, indicating that mTORC1 and Akt signal independently, thus explaining the phenotypic differences of NK cells stimulated in the presence of Akt vs. mTORC1 inhibitors[1]. These results are in line with our previous study in IL-2-cultured CTLs, which

showed mTORC1 signalling was unaffected following pharmacological or transgenic inhibition of PI3K/Akt signalling[12]. Instead, the data in CTL and now in NK cells argue that IL-2 regulation of leucine transport into the cell via SLC7A5 is a key mechanism in promoting mTORC1 signalling[40]. While the data here show that Akt signalling is not required for cytokine-induced NK cell metabolism and function, this does not preclude a role for PI3K/Akt signalling for NK cell metabolic responses in a different context such as in receptor activated NK cells. Indeed, PI3K has been shown to be important for NK cell receptor-mediated IFNγ production[45].

While the data show that glutamine is required for sustaining cMyc expression, it is not an important fuel for cytokine-stimulated NK cells. Glutamine does feed into the TCA cycle through glutaminolysis, but this metabolic pathway is a minor contributor to OXPHOS rates. Instead, the citrate malate shuttle, a glucose-fuelled metabolic pathway recently described in IL-2/IL-12-activated NK cells[2], is the main pathway responsible for sustaining elevated rates of OXPHOS. A key signalling molecule involved in the control of the citrate malate shuttle is the Srebp transcription factor (Sterol element binding protein)[2]. Taken together, cMyc and Srebp are two key factors in determining metabolic responses in IL-2/IL-12-stimulated NK cells; cMyc promotes glycolysis and mitogenesis while Srebp controls the metabolic switch to the citrate malate shuttle to fuel OXPHOS.

NK cells isolated from human solid tumours have been found to be defective in their pro-inflammatory functions including IFNγ production and tumour cytotoxicity[5–7]. Tumour cells are known to have a high demand for glutamine, in addition to glucose, and so it is likely that the tumour microenvironment can also have low levels of glutamine[46,47]. The data presented here argue that glutamine-restricted tumour microenvironments will inhibit cMyc expression in NK cells, leading to reduced NK cell metabolism and the inhibition of anti-tumour NK cell functions. Indeed, the data show that when IL-2/IL-12-activated NK cells were switched into glutamine-deficient conditions, metabolic rates of OXPHOS and glycolysis decreased dramatically and IFNγ production and tumour cytotoxicity was substantially inhibited. Given that glutamine is a crucial fuel for tumour cells, targeting glutamine metabolism is an attractive anti-cancer strategy that is being actively pursued[46,48]. However, it is important to consider the likely impact of such strategies on the anti-tumour immune response. Anti-cancer drugs targeting glutamine metabolism will also inhibit the growth, proliferation and function of T cells[14,49]. However, this study reveals that NK cell metabolism and functional responses are not affected by inhibitors of glutamine metabolism. This finding has positive implications for the efficacy of anti-cancer therapies using inhibitors of glutamine metabolism. In fact, it might be predicted that such inhibitors might lead to increased levels of glutamine within the tumour microenvironment, due to decreased utilisation by tumour cells, which would facilitate the expression of cMyc and anti-tumour effector functions in tumour-infiltrating NK cells. This will be an exciting area to address in future studies.

## Methods

**Mice.** C57BL/6J male mice were purchased from Harlan (Bicester, UK) or were bred in house. Mice with loxP sites inserted flanking exon 2 of the *Myc* gene (B6.129S6-Myc$^{tm2Fwa}$/Mmjax)[50], mice with loxP sites inserted flanking exon 2 of the *Hif1a* gene (B6.129−Hif1a$^{tm3Rsjo}$)[51], mice with loxP sites inserted flanking exon 1 of the *Slc7a5* gene (B6.129P2-Slc7a5$^{tm1.1Daca}$/J)[40], mice in which a fusion protein of Myc and enhanced green fluorescent protein is expressed in exon 2 of the *Myc* gene (B6;129-Myc$^{tm1Slek}$/J)[52] and transgenic mice expressing cre recombinase under the control of the vav promoter (B6.CgTg(Vavl-cre)A2 Kio/J)[53] were from The Jackson Laboratory. Transgenic mice expressing a tamoxifen inducible cre-recombainse (Gt(ROSA)26Sor<tm2(cre/ERT2) Brn/Cnrm)[54] were obtained from the European Mouse Mutant Archive (EMMA). All mice used in this study were between 8 and 12 weeks of age and were bred and maintained in compliance with

EU and the Health Products Regulatory Authority regulations with the approval of the University of Dublin's ethical review board and in accordance with the WTB/RUTG, University of Dundee in compliance with UK Home Office Animals (Scientific Procedures) Act 1986 guidelines.

**Cell culture.** Cells were cultured in RPMI media (Invitrogen) supplemented with 2 mM L-glutamine (Invitrogen), 10% heat-inactivated fetal calf serum (FCS; Labtech International), 50 μM β-Mercaptoethanol (Sigma) and 1% penicillin/streptomycin (Invitrogen/Biosciences). For western blot analysis in amino acid-deficient media, RPMI media without the amino acids L-glutamine and L-leucine (MP Bio) were supplemented with 10% heat-inactivated, dialysed FCS (Labtech International), 1% penicillin/streptomycin (Invitrogen/Biosciences) and 50 μM β-mercaptoethanol. For the amino acid-deficient stimulation, either 2 mM L-glutamine (Invitrogen), 0.38 mM L-leucine (Sigma) or both amino acids together with rapamycin (20 nM, Fisher) was added to the media. For ex vivo NK cell experiments, splenocytes were isolated from murine spleens and left unstimulated with low-dose IL-15 (5 ng/ml) as a survival factor or were activated with IL-2 (20 ng/ml, NCI preclinical repository) and IL-12 (10 ng/ml, Miltenyi Biotech) for 18 h. For NK cell culture, splenocytes were isolated from the murine spleen and cultured in IL-15 (10 ng/ml, Peprotech; in RPMI media) for 4 days. On day 4, the cells were supplemented with IL-15 (10 ng/ml) and cultured for a further 2 days. On day 6, the cells were stimulated for 18 h with IL-2 (20 ng/ml) and IL-12 (10 ng/ml) or were stimulated for 20 h with IL-2 (20 ng/ml) and IL-12 (10 ng/ml) and were cultured for further 20 h in media containing IL-2 (20 ng/ml) and IL-12 (10 ng/ml) where indicated. For AKT experiments, NK cells were stimulated with IL-2 and IL-12 for 18 h in the presence or absence of Akti-1/2 (2 μM, Sigma). For signalling analysis, 18 h IL-2/IL-12-activated NK cells were treated with Akti-1/2 (2 μM, Sigma), the proteasomal inhibitor MG132 (3 μM, Sigma) or the GSK3 inhibitor CT99021 (2 μM, Sigma) for 1 h or DMOG (200 μM, Sigma) for 2 h prior to protein lysis. The cells were cultured in the presence or absence of glutamine (2 mM) in RPMI supplemented with dialysed FCS, or alternatively in glutamine containing media plus the inhibitor DON (2 μM Sigma) or the glutaminase inhibitor BPTES (10 μM). Unstimulated cells were maintained in low-dose IL-15 (5 ng/ml) as a survival factor. For SLC7A5 inhibition experiments, the concentration of amino acids in RPMI was diluted twofold using Hank's balanced salt solution (HBSS; Invitrogen) in the presence or absence of BCH (25 mM Sigma). For biochemical analyses, NK cells were purified by magnetic-activated cell sorting (MACS) using the NK cell isolation kit II (Miltenyi Biotech) from the culture after day 6 or directly ex vivo. Where indicated, splenocytes isolated from *cMyc*$^{−/−}$ (*cMyc*$^{flox/flox}$ × Tamox-cre) or WT (*cMyc*$^{WT/WT}$ × Tamox-cre) mice were cultured for 4 days in IL-15 (10 ng/ml, Peprotech; in RPMI media) in the presence of 4-hydroxytamoxifen (0.6 μM, Sigma) to induce cre recombinase-mediated excision of the floxed cMyc exon. The 4-hydroxytamoxifen (0.6 μM, Sigma) was re-added on day 4 when cultures were fed with IL-15 (10 ng/ml, Peprotech). For T-cell culture splenocytes were isolated from murine spleen and T cells were activated with anti-CD3 antibody (2c11, 500 ng/ml) and IL-2 (20 ng/ml, NCI preclinical repository) in RPMI media for 36 h at 10 × 10⁶ cells/ml. Following activation, cells were washed and maintained in IL-2 (20 ng/ml). IL-2 (20 ng/ml) was re-added for the entire culture volume and the cell concentration adjusted to 0.3 × 10⁶ cells/ml every 24 h for a further 3 days before being put in the presence or absence of glutamine (2 mM) in RPMI supplemented with dialysed FCS, or alternatively in glutamine containing media plus DON (2 μM) for a further 3 days. For proliferation analysis of CD4⁺ T cells, T cells isolated from the murine spleen were stained with carboxyfluorescein succinimidyl ester (CFSE; BD Biosciences) before being activated with anti-CD3 antibody (2C11, 1 μg/ml, BD Biosciences) in RPMI media in the presence or absence of 10 μM BPTES, or maintained in IL-7 (10 μg/ml) for 36 h at 10 × 10⁶ cells/ml. After activation, cells were washed out of media and maintained in RPMI with IL-2 (20 ng/ml) for additional 48 h in the presence or absence of DON (2 μM) inhibitor. Proliferation was assessed by cell counts after 48 h. No significant differences in cell survival were observed as determined by flow cytometry analysis. For analysis of cytotoxicity, YAC-1 cells (mouse lymphoma cell line) and K562 cells (human chronic myelogenous leukaemia cell line) were cultured in either RPMI 1640 (Invitrogen/Biosciences) or Iscove's modified Dulbecco's medium (Sigma), both supplemented with 2 mM L-glutamine (Invitrogen, Biosciences), 10% heat-inactivated FCS (Labtech, International) and 1% penicillin/streptomycin (Invitrogen/Biosciences). YAC-1 and K562 cell lines were purchased from the American Type Culture Collection.

**Flow cytometry.** Cells were incubated for 10 min at 4 °C with Fc blocking antibody CD16/CD32 (2.4G2) and subsequently stained for 20 min at 4 °C with saturating concentrations of fluorophore conjugated antibodies. Antibodies used were as follows: NK1.1–eFluor 450 (PK136), NK1.1–BV421 (PK136), NKp46–PerCP eFluor 710 (29A1.4), NKp46–PE (29A1.4), CD3–FITC (145-2C11), TCRβ–APC (H57-597), CD69–PerCp-Cy5.5 (H1.2F3), CD25–APC-Cy7 (PC61), CD71–PE (C2), IFNγ–APC (XMG1.2), granzyme B–PE-Cy7 (NGZB), CD4–BV421 (GK1.5), CD19–PE-Cy7 (1D3), CD8a–PerCp-Cy5.5 (53-6.7), CD45.1–PerCpCy5.5 (A20), CD45.2–AlexaFlour 700 (A20), TCRb–PeCy7 (H57-597), CD69–PE (H1.2F3), NK1.1–APC (PK136) purchased from eBiosciences, Biolegend and BD Biosciences. Zombie Aqua™ (Biolegend) was used as a viability dye. Live cells were gated according to their forward scatter (FSC-A) and side scatter or according to Zombie Aqua™ negative cells, single cells according to their FSC-W and FSC-A, NK cells

were identified as NK1.1$^+$, NKp46$^+$ and CD3$^-$ cells and CD4$^+$ T cells were identified as CD19$^-$, NKp46$^-$, TCRβ$^+$ and CD4$^+$. For intracellular staining the cells were incubated for 4 h with the protein transport inhibitor GolgiPlug$^{TM}$ (BD Biosciences). For fixation and permeabilization of the cells, the Cytofix/Cytoperm kit from BD Biosciences was used according to manufacturer's instructions. Data were acquired on either a FACSCanto, a LSR Fortessa, or a FACSCalibur (Becton Dickinson) and analysed using FlowJo software (TreeStar). The gating strategies used are outlined in Supplementary Figure 8. For proliferation studies, splenocytes were stained with CFSE (BD Biosciences) according to the manufacturer's instruction. T cells were either maintained in IL-7 (10 μg/ml) or activated with anti-CD3 antibody (2C11, 1 μg/ml, BD Biosciences) in the presence or absence of 10 μM BPTES and were analysed on day 2 and day 3 by dilution of the dye CFSE. To determine cMyc expression CD45.1 WT splenocytes were mixed with CD45.2 GFP-Myc splenocytes in a 1:1 ratio prior to culture as described. The ratio of fluorescence in CD45.2 GFP-Myc and CD45.1 WT NK or T cells was calculated, the WT cells providing a internal control for autofluorescence.

**NK cell cytotoxicity.** For the measurement of NK cell cytotoxicity, IL-2 (20 ng/ml) and IL-12 (10 ng/ml) activated NK cells were stimulated for 18 h in RPMI medium containing dialysed FCS with IL-2 (20 ng/ml) plus IL-12 (10 ng/ml) with or without glutamine (2 mM) or the inhibitors DON (2 μM) or BPTES (10 μM). YAC-1 or K652 target cells were stained with 20 μM calcein-AM (BD Pharmingen) at 37 °C for 30 min. $3 \times 10^4$ stained target cells were added to a 96-well V-bottomed plate, and NK cells were added in a NK cell/target cell ratio of 5:1, 2:1 or 1:1. For measurement of spontaneous release and maximal release, only target cells were added to the well or a final concentration of 0.2% Triton X-100 lysis buffer was added to target cells, respectively. Cells were incubated for 4 h at 37 °C. After incubation, the plate was spun down at $200 \times g$ for 5 min, and 75 μl supernatant was transferred to a black 96-well plate. Fluorescence was measured on a Molecular Probes Spectra Max M3 spectrometer with an excitation wavelength of 495 nm and an emission wavelength of 515 nm. All samples were measured in triplicate, and the average was used for further analysis.

**Stable isotope tracer analysis.** For analysis of TCA cycle intermediates, $4 \times 10^6$ to $5 \times 10^6$ purified cultured NK cells were stimulated for 18 h in medium containing 2 mM unlabelled glutamine. For time course experiments, purified cultured NK cells were stimulated for 16 h in medium containing 2 mM unlabelled glutamine and IL-2 (20 ng/ml) plus IL-12 (10 ng/ml). The cells were washed three times in medium without glutamine, then $4 \times 10^6$ to $5 \times 10^6$ NK cells were cultured for another 30 min to 8 h in medium containing IL-2 (20 ng/ml) plus IL-12 (10 ng/ml) and 2 mM $^{13}C_5$-glutamine (purity 99.1%, CK Isotopes Limited). For all experiments, cells were harvested and washed three times with ice-cold phosphate-buffered saline (PBS). TCA cycle intermediates were extracted from the cell pellet and cell culture supernatant by adding 80% aqueous methanol. Extracts were dried using a centrifugal evaporator (GeneVac EZ-2, SP Scientific) and were stored at −80 °C until further processing. Cell pellets from the experiment with unlabelled glucose were spiked with 10 μl internal standard solution containing [U-$^{13}$C] fumarate, [U-$^{13}$C] succinate, malate-d3 and citrate-d4 at 1 mM each. Calibration curves for each analyte with the corresponding stable isotope labelled analogue as internal standard were used to obtain quantitative data.

**GC-MS analysis.** To analyse intracellular TCA cycle intermediates, the cell lysates were dried (CombiDancer Hettich AG) and subjected to methoximation and silylation employing the derivatization protocol and instrumental setup previously described[55]. Then, 1 μl of the derivatized sample was injected using splitless injection mode. Gas chromatography–mass spectrometry (GC-MS) data were analysed using the software Mass Hunter (version B.07.01/Build 7.1.524.0). All raw data were corrected for natural abundance and tracer impurity using the software IsoCor (Software Version 1.0)[56] for the GC-MS data. For comparison of the label incorporation of several metabolites, the mean isotopic enrichment is presented.

**Proteomics analysis.** For proteomic analysis, $5 \times 10^6$ purified cultured NK cells were stimulated for 18 h in RPMI media containing IL-2 (20 ng/ml) plus IL-12 (10 ng/ml). To remove dead cells, a density gradient (Lymphoprep, Axis-Shield) was used. Cells were spun down and stored at −80 °C until further preparation. Cell pellets were lysed in 400 μl lysis buffer (4% SDS, 50 mM TEAB pH 8.5, 10 mM TCEP). Lysates were boiled and sonicated with a BioRuptor (30 cycles: 30 s on, 30 s off) before alkylation with iodoacetamide for 1 h at room temperature in the dark. The lysates were subjected to the SP3 procedure for protein clean-up[47] before elution into digest buffer (0.1% SDS, 50 mM TEAB pH 8.5, 1 mM CaCl$_2$) and digested with LysC and Trypsin, each in a 1:50 (enzyme:protein) ratio. Tandem mass tag (TMT) labelling and peptide clean-up were performed according to the SP3 protocol. Samples were eluted into 2% dimethyl sulphoxide in water, combined and dried in vacuo. The TMT samples were fractionated using off-line high pH reverse phase chromatography: samples were loaded onto a $4.6 \times 250$ mm Xbridge$^{TM}$ BEH130 C18 column with 3.5 μm particles (Waters). Using a Dionex BioRS system, the samples were separated using a 25 min multistep gradient of solvents A (10 mM formate at pH 9 in 2% acetonitrile) and B (10 mM ammonium formate pH 9 in 80% acetonitrile) at a flow rate of 1 ml/min. Peptides were separated into 48

fractions which were consolidated into 24 fractions. The fractions were subsequently dried and the peptides redissolved in 5% formic acid and analysed by liquid chromatography–mass spectrometry (LC-MS).

**Liquid chromatography electrospray tandem mass spectrometry.** 1 μg per fraction was analysed using an Orbitrap Fusion Tribrid mass spectrometer (Thermo Scientific) equipped with a Dionex ultra high-pressure liquid chromatography system (nano RSLC). Reversed-phase liquid chromatography (RP-LC) was performed using a Dionex RSLC nano HPLC (Thermo Scientific). Peptides were injected onto a 75 μm × 2 cm PepMap-C18 pre-column and resolved on a 75 μm × 50 cm RP-C18 EASY-Spray temperature-controlled integrated column-emitter (Thermo) using a 4 h multistep gradient from 5% B to 35% B with a constant flow of 200 nl/min. The mobile phases were: 2% acetonitrile (ACN) incorporating 0.1% formic acid (FA; Solvent A) and 80% ACN incorporating 0.1% FA (Solvent B). The spray was initiated by applying 2.5 kV to the EASY-Spray emitter and the data were acquired under the control of Xcalibur software in a data-dependent mode using top speed and 4 s duration per cycle, and the survey scan is acquired in the Orbitrap covering the $m/z$ range from 400 to 1400 Th with a mass resolution of 120,000 and an automatic gain control (AGC) target of 2.0−e5 ions. The most intense ions were selected for fragmentation using collision-induced dissociation (CID) in the ion trap with 30% CID collision energy and an isolation window of 1.6 Th. The AGC target was set to 1.0−e4 with a maximum injection time of 70 ms and a dynamic exclusion of 80 s. During the MS3 analysis for more accurate TMT quantifications, 10 fragment ions were co-isolated using synchronous precursor selection using a window of 2 Th and further fragmented using HCD collision energy of 55%. The fragments were then analysed in the Orbitrap with a resolution of 60,000. The AGC target was set to 1.0−e5 and the maximum injection time was set to 300 ms.

**Database searching and reporter ion quantification.** The data were processed, searched and quantified with the MaxQuant software package, version 1.5.7.4. Proteins and peptides were identified using the UniProt mouse reference proteome database (SwissProt and Trembl accessed on 13.01.2017) and the contaminants database integrated in MaxQuant using the Andromeda search engine[48,49] with the following search parameters: carbamidomethylation of cysteine and TMT modification on peptide N-termini and lysine side chains were fixed modifications, while methionine oxidation, acetylation of N-termini of proteins, conversion of glutamine to pyro-glutamate and phosphorylation on STY were variable modifications. The false discovery rate was set to 5% for positive identification of proteins and peptides with the help of the reversed mouse Uniprot database in a decoy approach. Copy numbers were calculated as previously described[50] after allocating the summed MS1 intensities to the different experimental conditions according to their fractional MS3 reporter intensities.

**Quantitative real-time PCR.** Cultured NK cells were purified using MACS purification with the NK isolation kit II (Miltenyi Biotech) prior to stimulation. RNA was isolated using the RNeasy RNA purification mini kit (QIAGEN) according to the manufacturer's protocol. From purified RNA, complementary DNA (cDNA) was synthesised using the reverse-transcriptase kit qScript cDNA synthesis kit (Quanta Biosciences). Real-time PCR was performed in triplicate in a 96-well plate using iQ SYBR Green-based detection on an ABI 7900HT fast qPCR machine. For the analysis of mRNA levels the derived values were normalised to Rplp0 mRNA levels. Primer sequences are provided in Supplementary Table 1.

**Seahorse metabolic flux analyser.** For real-time analysis of the extracellular acidification rate (ECAR) and oxygen consumption rate (OCR) of purified and expanded NK cells cultured under various conditions, a Seahorse XF-24 Analyser, a Seahorse XFe-96 Analyser or a Seahorse XF-8 Analyser (Seahorse Bioscience) was used. In brief, 500,000 to 750,000 MACS purified, expanded NK cells were added to a 24-well XF Cell Culture Microplate, 100,000 to 200,000 MACS purified NK cells to a 96-well XFe Cell Culture Microplate and 200,000 ex vivo pure NK cells to an 8-well XF Cell Culture Microplate (Seahorse Biosciences). All cell culture plates were treated with Cell-Tak$^{TM}$ (BD Pharmingen) to ensure that the NK cells adhere to the plate. Sequential measurements of ECAR and OCR following addition of the inhibitors (Sigma) oligomycin (2 μM), FCCP (1 μM), rotenone (100 nM) plus antimycin A (4 μM), and 2-deoxyglucose (2DG, 30 mM) allowed for the calculation of basal glycolysis, glycolytic capacity, basal mitochondrial respiration, and maximal mitochondrial respiration. Where indicated, BMS303141 (10 μM, Sigma), SB204990 (30 μM, Tocris), BPTES (10 μM,Tocris) or an equivalent amount of vehicle control was injected into the Seahorse plate.

**Western blot analysis.** For western blot analysis, cells were harvested, washed twice with ice-cold PBS and lysed at $1 \times 10^7$/ml in lysis buffer containing 50 mM Tris Cl pH 6.7, 2% SDS, 10% glycerol, 0.05% Bromphenol Blue, 1 μM dithiothreitol, phosphatase- and protease inhibitors. Samples were denatured at 95 °C for 10 min, separated by sodium dodecyl sulphate–polyacrylamide gel electrophoresis and transferred to a polyvinylidene difluoride membrane. Antibodies used for probing blots are outlined in Supplementary Table 2. Uncropped western blot images are included in Supplementary Fig. 9.

**Amino acid uptake measurements**. For analysis of amino acid uptake, $1 \times 10^6$ NK cells were resuspended in 0.4 ml uptake medium. HBSS (Gibco) containing [$^3$H]L-phenylalanine (1 μCi/ml) and [$^{14}$C]L-glutamine (0.2 μCi/ml) was used for the analysis of phenylalanine and glutamine uptake. The uptake assay was done in the presence or absence of 10 mM BCH to inhibit System L–specific activity. Uptake was assayed for 3 min with samples layered over 0.5 ml of a mixture of silicone oil (Dow Corning 550 (BDH silicone products); specific density, 1.07 g/ml) and dibutyl phthalate (Fluka) in a ratio of 1:1. Cells were pelleted below the oil, then the aqueous supernatant solution, followed by the silicon oil–dibutyl phthalate mixture, was aspirated, and the cell pellet underneath was resuspended in 200 μl NaOH (0.5 M). Then, β-radioactivity was measured by liquid scintillation counting in a Beckman LS 6500 Multi-Purpose Scintillation Counter (Beckman Coulter).

**Statistical analysis**. GraphPad Prism 6.00 (GraphPad Software) was used for statistical analysis. A one-way or a two-way analysis of variance (ANOVA) with the Tukey or Sidak post hoc test was used for multiple comparisons. Student's *t*-test was used when there were only two data sets for comparison. For comparison of relative values, a one-sample *t*-test was used to calculate *p* values with the theoretical mean set to 1.00. A *p*-value of <0.05 was considered as statistically significant.

**Data availability**. All relevant data are available from the corresponding author.

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

## Acknowledgements

We are grateful for the facilities provided by the Comparative Medicine Unit, Trinity College Dublin. This work was supported by the Science Foundation Ireland (12/IP/1286 and 13/CDA/2161) and Marie Skłodowska-Curie Actions (PCIG11-GA-2012-321603), German Research Foundation (KFO262). K.L.O. is supported by an Irish Cancer Society Research Scholarship (CRS15OBR) including support from the Children's Leukaemia Research Project.

## Author contributions

D.K.F. and R.M.L. conceptualised the project; R.M.L., N.A., N.K.-M., L.V.S., K.L.O., K.D., J.L.H., C.M.G. and A.G. performed the experiments. D.K.F. wrote the original draft of the manuscript. D.K.F., R.M.L., C.M.G. and L.V.S reviewed and edited the manuscript. L.V.S. and P.J.O. provided resources essential to this study. D.K.F., D.A.C., L.V.S. and C.M.G. supervised the research.

## Additional information

**Competing interests:** The authors declare no competing interests.

