## [Peer Review File · Nature Communications]

Reviewers' comments:

Reviewer #1 (Remarks to the Author):

In this manuscript Loftus et al investigate the regulation of cMyc in NK cells and how this affects NK cell function. In a set of in vitro experiments they investigate regulation of cMyc after IL-2/12 stimulation and conclude that this cytokine stimulation upregulates mTORC1 signaling short-term which controls cMyc and long-term upregulates the Slc7a5 transporter which exports glutamine to regulate the synthesis of reduce degradation of cMyc. Overall, the hypothesis that glutamine is not utilized for metabolism by cytokine-activated NK cells but is required for transporter activity is intriguing, but there are concerns regarding evidence for this model as outlined below.

1. The authors use an inhibitor of glutamine metabolism, DON, to conclude that glutaminolysis is not required for the effects of glutamine on cMyc expression and NK cell function. The data in Figure 7e showing a minimal shift in GlcNAc by flow with inhibitor treatment are not convincing. In addition to inhibitor studies, a more direct measurement of glutamine metabolism (or lack thereof) and/or evidence it is being exported into the media would be more convincing. In addition, the data/explanation for differences in metabolism with 20hr vs. 1hr glutamine deprivation (Fig. S2 vs. 7) are not clear. Are the authors concluding that the effect of 20hr glutamine deprivation is due to lack of cMyc? If that is the case what was the effect on metabolism of culture of cMyc^{-/-} NK cells in glutamine free media for 20hr? What is the metabolism of wt NK cells treated with DON for 20hr (Fig 7c shows 1hr treatment only)?

2. BCH inhibits L amino acid transporter and both LAT1 (SLC7A5/SLC3A2) and LAT2 (SLC7A8/SLC3A2). Were other amino acid transporters measured? Why do the authors conclude that BCH inhibition is evidence of the importance of Slc7a5?

3. Why were the NK cells all cultured in IL-15 for 6 days prior to assays? This culture time and condition will fundamentally change the metabolism of the cells. Do freshly isolated NK cells upregulate cMyc with IL2/12 stimulation? Are these findings relevant to NK cells in vivo? At a minimum, experiments in Figures 1 & 2 establishing IL-2/12 upregulation of Myc and alterations in Myc⁻ NK cells should be completed with fresh NK cells (can use tamoxifen injected mice rather than culture). The text should also include whether fresh or IL-15 cultured NK cells were being used for each experiment, they are not the same. Since we know that the metabolism of IL-15-expanded NK cells is different from that of naïve NK cells, this is an important point when considering how these findings can be translated to the functional consequences of metabolic alterations in patients.

4. What was the phenotype and viability of the Myc⁻ NK cells after tamoxifen treatment? Were a population of NK cells lost when Myc was deleted, e.g., were mature or immature NK cells preferentially affected. Why weren't these experiments done in the mouse with tamoxifen treatment?

5. NK cell killing (figure 7I) should be expressed as % specific lysis rather than % target dead cells to account for non-specific cell death. In this assay it is odd that lower E:T ratios leads to increased killing, is there an error in the labeling? Overall the differences in killing are relatively low and a wider range of E:T ratios may demonstrate this better.

6. The schema figure shows IL-2 upregulating Slc7a5 – but the experimental evidence for this was with IL-2 + IL-12. What is the evidence that IL-2 upregulates this receptor?

7. The title should include "murine".

Minor:

1. Figure 1e, an isotype control should be shown to know whether cells are positive for CD71.
2. on p. 6 of the text, Supp Fig 2 is mislabeled as Supp fig 5.
3. The figure legend for Supplemental Figure 4 has multiple spelling errors and is missing references (reads as “[refs]”).
4. Additional background on cMyc and known roles in NK/lymphocyte biology should be included in the text.
5. A previous study showed no effect on NK cell IFN-g production with glutamine deprivation in fresh NK cells (Keppel, JI 2015). Why do the authors suspect results here are different – does this have to do with the difference between fresh and IL-15 expanded/activated NK cells?

Reviewer #2 (Remarks to the Author):

The manuscript by Loftus et al. reports the role of glutamine and Myc in the IL2/IL12-mediated activation of NK cell functional responses. The authors pursued the role of glutamine in NK cell activation as a follow up to their previous documentation of a role for glycolysis in NK cell activation. In the current manuscript, the authors provide experimental results that are construed to indicate that: 1) IL2/IL12-mediated activation of NK cells (herein ‘activation of NK cells’) requires glutamine, 2) glutamine withdrawal diminishes Myc protein expression and activation of NK cells, 3) Myc is necessary, but not HIF-1, for activation of NK cells, 4) the transporter SLC7A5, which is important for amino acid transport -, particularly as a glutamine-leucine antiporter, is inferred to be required for activation of NK cells via the use of a non-specific inhibitor BCH, 5) Myc levels are dependent on mTORC1 5) GSK3beta is involved in diminished Myc levels after glutamine withdrawal, 6) inhibition of glutamine metabolism via a non-specific glutaminase inhibitor DON did not affect activation of NK cells but did affect T cell activation. Overall, this is a potentially important contribution to the literature particular in the emerging area of immunometabolism and NK cell function; however, there are many technical issues that should be addressed.

1. In T cells, Myc is required early after stimulation, but HIF-1 appears to be required for sustained T cell differentiation. The authors should address or comment on whether HIF-1 is required for long term function of NK cells. Specifically, there is no in vivo functional studies specifically targeting NK cells’ Myc or HIF status (by adoptive transfer, for example).
2. The use of Rapamycin to probe mTOR is shown, but the data could be significantly enhanced with TOR kinase inhibitors.
3. Figure 4d: While inferring that mTORC1 is necessary for Myc expression, the authors found that ‘without leucine could not sustain mTORC1 signalling but these cells had normal levels of Myc.’ This appears to contradict the observation that Rapamycin diminished Myc protein levels in Figure 3i.
4. With the manipulations discussed in issues #2 and #3 (above), IFNg and Gzymb levels under various conditions should be shown so that Myc levels could be correlated with ‘function.’
5. The use of BCH should be accompanied by an abundance of caution, given that off-target effects cannot be ruled out. Further the authors did not addressed the possible roles of other transporters such as SLC1A5 or xCT, which are also involved in glutamine metabolism (see Altman et al. Nature Rev Cancer 2017).
6. Figure 7. DON is a structural analog of glutamine and non-specific inhibitor of glutaminase, Gls. As such an abundance of caution here is necessary as well. The use of BPTES as a specific inhibitor of Gls would directly address the issue confronting the authors’ query into glutamine anaplerosis. Given that DON could inhibit glutaminase as well as glucosamine synthesis, the non-specific nature

of DON makes the conclusion drawn by the authors questionable. There are also no metabolomics data to support any of the claims made on glutamine metabolism.

Reviewer #3 (Remarks to the Author):

The major findings here are that Myc is essential for IL2/IL12-induced metabolism and functional response and is regulated by glutamine. Glutamine withdrawal results in loss of cMyc impacting on NK cell biology.

The authors should be congratulated on a in depth investigation into a much needed area, NK cell metabolism.

I would like to know why IL-2 was chosen over the physiologically relevant IL-15? IL-2 induces a drastically different transcriptional profile and biological response in NK cells compared to IL-15. Given IL-15 would govern the early phase of an NK cell response to pathogen infection it would be helpful to explain why IL-2 was used here (besides the cost of IL-2 being significantly cheaper than IL-15).

The dashed lines indicating drug addition in the ECAR/OCR plots in figure 1 and 5 don't line up with the time point where the drug was added

Please find below our point by point responses to the reviewers concerns in red text.

Reviewer #1 (Remarks to the Author):

In this manuscript Loftus et al investigate the regulation of cMyc in NK cells and how this affects NK cell function. In a set of in vitro experiments they investigate regulation of cMyc after IL-2/12 stimulation and conclude that this cytokine stimulation upregulates mTORC1 signaling short-term which controls cMyc and long-term upregulates the Slc7a5 transporter which exports glutamine to regulate the synthesis of reduce degradation of cMyc. Overall, the hypothesis that glutamine is not utilized for metabolism by cytokine-activated NK cells but is required for transporter activity is intriguing, but there are concerns regarding evidence for this model as outlined below.

1. The authors use an inhibitor of glutamine metabolism, DON, to conclude that glutaminolysis is not required for the effects of glutamine on cMyc expression and NK cell function. The data in Figure 7e showing a minimal shift in GlcNAc by flow with inhibitor treatment are not convincing. In addition to inhibitor studies, a more direct measurement of glutamine metabolism (or lack thereof) and/or evidence it is being exported into the media would be more convincing. In addition, the data/explanation for differences in metabolism with 20hr vs. 1hr glutamine deprivation (Fig. S2 vs. 7) are not clear. Are the authors concluding that the effect of 20hr glutamine deprivation is due to lack of cMyc? If that is the case what was the effect on metabolism of culture of cMyc^{-/-} NK cells in glutamine free media for 20hr? What is the metabolism of wt NK cells treated with DON for 20hr (Fig 7c shows 1hr treatment only)?

- We would like to thank reviewer 1 for the insightful and constructive comments on our manuscript. The reviewer requests additional evidence that glutamine is not an important fuel in cytokine activated NK cells. We now include additional data using a specific inhibitor of glutaminase, BPTES, that confirms our original observations using the general inhibitor of glutamine metabolism DON. Additionally, we have performed ¹³C₅Gln in cytokine activated NK cells and additional experiments using the seahorse extracellular flux analyser. These experiments have been very informative in that they show that while glutamine does feed into the TCA cycle, glutaminolysis is of minor importance in sustaining OXPHOS levels. Our recent publication in Nat. Immunol (Assmann et al, 2017 Nov;18(11):1197-1206) showed that a glucose fueled citrate malate shuttle is crucial in the fueling of OXPHOS in IL2/IL12 stimulated NK cells. Indeed, we now include data directly comparing the contribution of the glucose fueled citrate malate shuttle and the glutamine fueled TCA to OXPHOS rates. Considering these new data, we have amended the manuscript to say that glutamine is not an important fuel for driving OxPhos rather than saying that glutamine is not a fuel for NK cells.

- Yes, we would argue that the differences observed in metabolism comparing 1 h and 20 h Gln deprivation are due to the effects of losing cMyc expression. We now include additional data to support this argument. We activated NK cells for 20 hours and then switched these

cells into media without glutamine or media with glutamine plus the glutaminase inhibitor BPTES for a further 20 hours before metabolic analyses. Glutamine withdrawal results in loss of Myc within 30 min whereas BPTES does not affect Myc expression. Gln withdrawal resulted in dramatic decreases in OCR while BPTES had a much smaller impact on OCR. Crucially, Gln withdrawal but not BPTES resulted in a substantial decrease in glycolysis and glycolytic capacity (Figure 8c-f), which we have shown to be controlled by cMyc in NK cells (Figure 1).

2. BCH inhibits L amino acid transporter and both LAT1 (SLC7A5/SLC3A2) and LAT2 (SLC7A8/SLC3A2). Were other amino acid transporters measured? Why do the authors conclude that BCH inhibition is evidence of the importance of Slc7a5?

• We agree that BCH can actually inhibit the whole family of LAT transporters (LAT1-4). However, only LAT1 is expressed to an appreciable level in IL2/12 stimulated NK cells as demonstrated by quantitative proteomics data now included in the manuscript and also shown here (below) Therefore, BCH is specifically targeting LAT1 in our experiments. We have amended the text to make this clear in the text and abstract.

3. Why were the NK cells all cultured in IL-15 for 6 days prior to assays? This culture time and condition will fundamentally change the metabolism of the cells. Do freshly isolated NK cells upregulate cMyc with IL2/12 stimulation? Are these findings relevant to NK cells in vivo? At a minimum, experiments in Figures 1 & 2 establishing IL-2/12 upregulation of Myc and alterations in Myc- NK cells should be completed with fresh NK cells (can use tamoxifen injected mice rather than culture). The text should also include whether fresh or IL-15 cultured NK cells were being used for each experiment, they are not the same. Since we know that

the metabolism of IL-15-expanded NK cells is different from that of naïve NK cells, this is an important point when considering how these findings can be translated to the functional consequences of metabolic alterations in patients.

- We thank the reviewer for these comments and we appreciate the concern that the reviewer has regarding studying metabolism in freshly isolated NK cells versus IL15 expanded NK cells. The reason that we have used IL15 expanded NK cells is to facilitate the types of metabolic analyses that are required to accurately characterize the metabolic pathways utilized by NK cells (seahorse analysis, metabolomics, which require numbers of NK cells that cannot be practically obtained directly from freshly isolated splenocytes). NK cells expanded in low dose IL15 are not overtly activated and undergo robust metabolic and functional responses when stimulated with IL2/12 cytokine (see Donnelly et al 2014, J. Immunol Nov 1;193(9):4477-84). That said, we agree that it is important to show that NK cells isolated from splenocytes undergo the same metabolic responses and now include data confirming this to be case. The data shows that splenic NK cells stimulated with IL2/12 exhibit robust increases in both glycolysis and OXPHOS, as seen for IL15 expanded NK cells.

- We also confirmed that in IL2/12 stimulated splenic NK cells:
 - (1) there is an increase in cMyc expression that is prevented when Slc7a5 is inhibited with BCH.
 - (2) the robust IL2/IL12 induced increases in ECAR and OCR are prevented when Slc7a5 (and hence also cMyc expression) was inhibited with BCH.
 - (3) IL2/IL12 induced IFN γ production and Granzyme B expression are prevented when Slc7a5 (and hence also cMyc expression) was inhibited with BCH.

- In my experience of the tamoxifen inducible Cre system in T cells and NK cells, the deletion of target genes is quite inefficient and differs for each locus targeted. We carefully optimized and monitored cMyc deletion in our cultured NK cells and found that the efficiency of cMyc deletion was variable (only data where >90% cMyc mRNA knockdown was confirmed is included in the paper) and so we decided that in vivo cMyc knockdown using tamoxifen is not practical.

- We have included clarification on the use of freshly isolated NK cells versus cultured NK cells.

4. What was the phenotype and viability of the Myc- NK cells after tamoxifen treatment? Were a population of NK cells lost when Myc was deleted, e.g., were mature or immature NK cells preferentially affected. Why weren't these experiments done in the mouse with tamoxifen treatment?

- There were no differences in the viability of Myc KO NK cells compared to WT NK cells (see below). Also, the subpopulations of NK cells were present in equivalent ratios in Myc KO and WT NK cells; this data is now included in Supplementary Figure 1a).

- The efficiency of Tamoxifen induced excision of floxed alleles varies in different locus' (I have worked with 4 different Tamoc-Cre mouse models – cMyc, HIF1a, HIF1b, Lkb1). Due to

variability in cMyc knockout in IL15 cultured NK cells following tamoxifen treatment we decided that in vivo cMyc knockdown using tamoxifen treated mice was not practical. All data presented in this manuscript are from cultures where cMyc knockdown was confirmed to be >90% by rtPCR. Given the number of NK cells in the spleens of mice (~ 1 million/spleen), confirming cMyc knockout in NK cells from tamoxifen treated mice in every experiment would not be practical.

5. NK cell killing (figure 7l) should be expressed as % specific lysis rather than % target dead cells to account for non-specific cell death. In this assay it is odd that lower E:T ratios leads to increased killing, is there an error in the labeling? Overall the differences in killing are relatively low and a wider range of E:T ratios may demonstrate this better.

- We have now performed a significant number of additional killing assays using a number of different target cells. These new data show large and highly significant differences in target cell killing between conditions of glutamine deprivation and BPTES or DON (Figure 8k, Supplementary Figure 6g). The data has been expressed as % specific lysis.

6. The schema figure shows IL-2 upregulating Slc7a5 – but the experimental evidence for this was with IL-2 + IL-12. What is the evidence that IL-2 upregulates this receptor?

- We now include data showing that in NK cells stimulated for 20 hours with IL2/12, the withdrawal of IL2, but not of IL12, for 8 hours results in the loss of Slc7a5 mRNA expression (Figure. 4b).

7. The title should include “murine”.

- We have included murine in the title.

Minor:

1. Figure 1e, an isotype control should be shown to know whether cells are positive for CD71.

We thank the reviewer for highlighting this important point. We now include in supplementary figure 1b data including an FMO control for CD71 demonstrating that unstimulated NK cells are negative for CD71 expression. We have amended the text to say that unstimulated NK cells do not express CD71.

2. on p. 6 of the text, Supp Fig 2 is mislabeled as Supp fig 5.

• Apologies for this error. We have now corrected this mistake.

3. The figure legend for Supplemental Figure 4 has multiple spelling errors and is missing references (reads as “[refs]”).

• Apologies for these errors. We have now corrected these mistakes.

4. Additional background on cMyc and known roles in NK/lymphocyte biology should be included in the text.

We have included in the introduction some addition background on cMyc in lymphocytes.

5. A previous study showed no effect on NK cell IFN-g production with glutamine deprivation in fresh NK cells (Keppel, JI 2015). Why do the authors suspect results here are different – does this have to do with the difference between fresh and IL-15 expanded/activated NK cells?

• We would like to thank the reviewer for this comment. We would like to highlight that our data presented herein and that of Keppel et al (2015) are not contradictory and explain why we have come to this conclusion.

Keppel et al performed 2 types of experiment with glutamine free conditions:

- (1) They stimulated fresh splenic NK cells for 6 hours with cytokine IL12/18 in complete media or with media lacking glutamine but also containing 2 additional metabolic inhibitors. There was no effect on IFNg in this experiment. This experiment is not comparable to our study as they inhibited multiple metabolic pathways simultaneously and because 6 hours is not sufficient time for cMyc dependent metabolic reprogramming to occur.
- (2) They activated NK cells with high dose IL15 for 72 hours (therefore these cells were not fresh splenic NK cells), then restimulated these cells with anti-NK1.1 for 6 hours +/- glutamine; no differences in IFNg were observed. In contrast, we deprived our IL2/IL12 activated NK cells of glutamine for 18 hours before effects on IFNg, GnzB and killing were observed. We also have data for NK cells deprived of glutamine for 6

hours where there is no impact upon IFNg production (data not shown).

Reviewer #2 (Remarks to the Author):

The manuscript by Loftus et al. reports the role of glutamine and Myc in the IL2/IL12-mediated activation of NK cell functional responses. The authors pursued the role of glutamine in NK cell activation as a follow up to their previous documentation of a role for glycolysis in NK cell activation. In the current manuscript, the authors provide experimental results that are construed to indicate that: 1) IL2/IL12-mediated activation of NK cells (herein 'activation of NK cells') requires glutamine, 2) glutamine withdrawal diminishes Myc protein expression and activation of NK cells, 3) Myc is necessary, but not HIF-1, for activation of NK cells, 4) the transporter SLC7A5, which is important for amino acid transport - , particularly as a glutamine-leucine antiporter, is inferred to be required for activation of NK cells via the use of a non-specific inhibitor BCH, 5) Myc levels are dependent on mTORC1 5) GSK3beta is involved in diminished Myc levels after glutamine withdrawal, 6) inhibition of glutamine metabolism via a non-specific glutaminase inhibitor DON did not affect activation of NK cells but did affect T cell activation. Overall, this is a potentially important contribution to the literature particular in the emerging area of immunometabolism and NK cell function; however, there are many technical issues that should be addressed.

We would like to thank the reviewer for these positive comments on our manuscript and we trust that we have addressed the technical issues identified in the revised manuscript.

1. In T cells, Myc is required early after stimulation, but HIF-1 appears to be required for sustained T cell differentiation. The authors should address or comment on whether HIF-1 is required for long term function of NK cells. Specifically, there is no in vivo functional studies specifically targeting NK cells' Myc or HIF status (by adoptive transfer, for example).

• In T cells cMyc is required early after TCR stimulation and then HIF1 α is induced once the T cells make IL2 and is required for sustained metabolic responses. The interesting point is the fact that IL2 promotes similar metabolic phenotypes in NK cells and cytotoxic T cells (CTL) through different mechanisms, cMyc in NK and HIF1 α in CTL. We have included proteomic data to show the relative expression of cMyc vs HIF1 α in these NK cells (8 fold more copies of cMyc protein to HIF1 α protein). Our data does not preclude an important role for HIF1 α in prolonged NK cells response in particular at sites of infection or within tumours where hypoxic conditions are likely. This has been discussed more extensively in the discussion.

2. The use of Rapamycin to probe mTOR is shown, but the data could be significantly enhanced with TOR kinase inhibitors.

- The reasoning behind using rapamycin is that rapamycin is the only inhibitor that specifically target the mTORc1 and not the second mTOR complex mTORC2. In contrast, mTOR kinase inhibitors inhibit both mTORC1 and mTORC2 kinases. While mTORC1 is known to have important roles in controlling lymphocyte metabolism, there is little evidence to suggest that mTORC2 plays an important role in the control of lymphocyte metabolism. In this study we have specifically addressed the role of mTORC1 and not mTORC2 in the control of cMyc and NK metabolism. We felt that the use of ATP competitive kinase inhibitors, inhibiting both mTORC1 and mTORC2, would complicate matters. We do agree that studying the relative roles for mTORC1 and mTORC2 would be of great interest but we feel that the only way to robustly address this question would be through the use of Raptor and Rictor KO mice to abolish mTORC1 and mTORC2, respectively, and this would be beyond the scope of this study.

3. Figure 4d: While inferring that mTORC1 is necessary for Myc expression, the authors found that ‘without leucine could not sustain mTORC1 signaling but these cells had normal levels of Myc.’ This appears to contradict the observation that Rapamycin diminished Myc protein levels in Figure 3i.

- We apologise for the confusion caused. We suggest that mTORC1 is only required for initial cMyc expression (min-hours) but not for sustained cMyc expression (18 hours). It appears that we included a cMyc blot in the original manuscript that was not a good representative image and was misleading. Our experiments (n=6) conclude that there is minimal effect of rapamycin on cMyc expression; NK cells stimulated with IL2/12 +/- rapamycin for 18 hours (See below). The image we included did not reflect this and we apologise for this. We have confirmed that 18 hour rapamycin treatment does not inhibit cMyc protein levels by proteomics analysis (See below, data for 3 separate experiments). Therefore, we would contend that the data with leucine deprivation and rapamycin is not contradictory. We have now included a more accurate representative figure showing cMyc protein levels in NK cells treated +/- rapamycin (Figure 3j).

- We also include additional data showing that NK cells stimulated for 18 hours with IL2/IL12 and then treated with rapamycin for 1 hour to not have decreased cMyc expression (Fig. 4k), further demonstrating that mTORC1 is not required for sustained cMyc expression in IL2/IL12 stimulated NK cells.

4. With the manipulations discussed in issues #2 and #3 (above), IFN γ and GrzymB levels under various conditions should be shown so that Myc levels could be correlated with ‘function.’

- As discussed above for #2, we believe the use of mTOR kinase inhibitors would complicate this current study as they also inhibit mTORC2.
- We have previously published that rapamycin treatment inhibits NK cell IFN γ production and granzyme b expression (Donnelly et al, JI 2014) but it should be noted that we have shown that mTORC1 regulates multiple metabolic pathways including the Srebp-regulated citrate-malate shuttle (Assmann et al, Nat. Immunol, 2017). So while inhibition of mTORC1 using rapamycin and leucine deprivation does not inhibit cMyc it does inhibit NK cell function through impacting these other metabolic pathways. In this present study we have

shown that Myc controls the rate of NK cell metabolism (glycolysis and OxPhos) and function. We now include additional data showing that NK cell function correlates with cMyc expression (Fig. 8). The data shows that Gln deprivation (leading to loss of cMyc) but not inhibition of glutamylolysis (cMyc protein levels unchanged) leads to decreased glycolytic rates after 20 hours, which correlate to decreased NK cell IFN γ production, granzyme b expression and tumour cell killing.

5. The use of BCH should be accompanied by an abundance of caution, given that off-target effects cannot be ruled out. Further the authors did not address the possible roles of other transporters such as SLC1A5 or xCT, which are also involved in glutamine metabolism (see Altman et al. Nature Rev Cancer 2017).

- We agree with the reviewer that any data generated using an inhibitor needs to be considered with care due to the possibility of off target effects. We now include additional data to provide additional confidence that the effects observed following the addition of BCH are due to the inhibition of Slc7a5:

- 1) BCH is a System L amino acid transporter (LAT) competitor and targets the 4 members of the LAT family (LAT1-4). We have amended the text and abstract to clarify this. We now include quantitative proteomics data showing that only Slc7a5 (LAT1) is expressed at appreciable levels in IL2/IL12 stimulated NK cells (Figure 4d). Therefore, in these NK cells BCH is specifically targeting Slc7a5/LAT1 in our experiments.

- 2) We have also been able to get some NK cell samples from mice that lack Slc7a5 expression in haematopoietic cells (Slc7a5 Vav-Cre mice). In these Slc7a5 deficient NK cells IL2/IL12 stimulation does not induce the expression of cMyc (Figure 4f).

- We agree that it is important to consider other transporters involved that might be involved in glutamine metabolism in our NK cells. We now include quantitative proteomics data showing that Slc1a5 and Slc38a2 are the major glutamine transporters in NK cells.
- Slc7a11 (also called xCT), which is linked to glutamine metabolism because it exports glutamate, is not expressed in NK cells based on the Immgen database and our proteomics dataset (data not shown).

6. Figure 7. DON is a structural analog of glutamine and non-specific inhibitor of glutaminase, Gls. As such an abundance of caution here is necessary as well. The use of BPTES as a specific inhibitor of Gls would directly address the issue confronting the authors' query into glutamine anaplerosis. Given that DON could inhibit glutaminase as well as glucosamine synthesis, the non-specific nature of DON makes the conclusion drawn by the authors questionable. There are also no metabolomics data to support any of the claims made on glutamine metabolism.

- We would like to thank the reviewer for this comment and suggestion. We agree that the use of a specific inhibitor of glutaminase is an important experiment. We have repeated our experiments with BPTES as suggested. These new data correlate to the experiments

performed with DON. We have now included the data obtained with BPTES (Figure 7-8) and DON (supplementary figure 5-6) in the revised manuscript.

We agree that it is important to include metabolomics data to support the argument that glutamine metabolism is not important in sustaining OXPHOS in IL2/IL12 stimulated NK cells. We have now performed ¹³C₅-glutamine metabolic tracing experiments, non-labelled metabolomics and additional seahorse experiments to investigate this question in detail. The data shows that while glutamine does feed into the TCA intermediates, the glutamine-fueled TCA cycle is a minor pathway for sustaining the elevated levels of OXPHOS in these activated NK cells. Instead, the glucose-fueled Citrate Malate shuttle is the primary pathway fuelling OxPhos in these NK cells (Figure 7e-h and Assmann et al., Nat Immunol, 2017, Nov;18(11):1197-1206. doi: 10.1038/ni.3838)

Reviewer #3 (Remarks to the Author):

The major findings here are that Myc is essential for IL2/IL12-induced metabolism and functional response and is regulated by glutamine. Glutamine withdrawal results in loss of cMyc impacting on NK cell biology.

The authors should be congratulated on a in depth investigation into a much needed area, NK cell metabolism.

I would like to know why IL-2 was chosen over the physiologically relevant IL-15? IL-2 induces a drastically different transcriptional profile and biological response in NK cells compared to IL-15. Given IL-15 would govern the early phase of an NK cell response to pathogen infection it would be helpful to explain why IL-2 was used here (besides the cost of IL-2 being significantly cheaper than IL-15).

We have also previously published on the metabolic changes induced by IL2/12 in murine NK cells and this current paper builds on these findings and provides a mechanism of action for the responses observed (1, 2). Changing cytokines for these experiments would change the biological context for the mechanistic experiment.

However, in terms of original cytokine choice, IL2 has been long recognised as an important cytokine in NK cell biology and is one of the potent activators of both human and murine NK cells. While IL15 is more physiologically relevant during NK cell development and is important in particular viral infections, there is also strong evidence that IL2 impacts on NK cell responses after the initial innate immune response e.g. IFN γ production by NK cells post-vaccination or post viral infection is dependent on T cell derived IL2 (3, 4). IL2 will also be produced rapidly by T cells during memory responses and be available for early activation of NK cells during the time frame generally associated as part of the innate immune response.

The dashed lines indicating drug addition in the ECAR/OCR plots in figure 1 and 5 don't line up with the time point where the drug was added.

• The drugs were injected in-between metabolic readings and so the dashed lines are not meant to line up with any exact timepoints. For instance, for Fig 1k the first injection occurs after the 2nd measurement at about 15-16 min.

1. Donnelly, R. P., R. M. Loftus, S. E. Keating, K. T. Liou, C. A. Biron, C. M. Gardiner, and D. K. Finlay. 2014. mTORC1-dependent metabolic reprogramming is a prerequisite for NK cell effector function. *J Immunol* 193: 4477-4484.
2. Assmann, N., K. L. O'Brien, R. P. Donnelly, L. Dyck, V. Zaiatz-Bittencourt, R. M. Loftus, P. Heinrich, P. J. Oefner, L. Lynch, C. M. Gardiner, K. Dettmer, and D. K. Finlay. 2017. Srebp-controlled glucose metabolism is essential for NK cell functional responses. *Nat Immunol* 18: 1197-1206.
3. Horowitz, A., R. H. Behrens, L. Okell, A. R. Fooks, and E. M. Riley. 2010. NK cells as effectors of acquired immune responses: effector CD4⁺ T cell-dependent activation of NK cells following vaccination. *Journal of immunology* 185: 2808-2818.
4. He, X. S., M. Draghi, K. Mahmood, T. H. Holmes, G. W. Kemble, C. L. Dekker, A. M. Arvin, P. Parham, and H. B. Greenberg. 2004. T cell-dependent production of IFN-gamma by NK cells in response to influenza A virus. *J Clin Invest* 114: 1812-1819.

REVIEWERS' COMMENTS:

Reviewer #1 (Remarks to the Author):

The authors have addressed all of the prior concerns and have performed additional experiments to support their hypotheses.

Reviewer #4 (Remarks to the Author):

The authors have sufficiently addressed the previous concerns.